# ⛏ MINEDRAFT: A Framework for Batch Parallel Speculative Decoding

**Zhenwei Tang** [1 2 *]  **Arun Verma** [2 *]  **Zijian Zhou** [2 3]  **Zhaoxuan Wu** [2]  **Alok Prakash** [2]
**Daniela Rus** [4 2]  **Bryan Kian Hsiang Low** [3 2]

## Abstract

Speculative decoding (SD) accelerates large language model inference by using a smaller draft model to propose draft tokens that are subsequently verified by a larger target model. However, the performance of standard SD is often limited by the strictly sequential execution of these drafting and verification stages. To address this, this paper proposes MINEDRAFT, a batch parallel speculative decoding (PSD) framework designed to effectively hide drafting latency by overlapping it with verification. Our theoretical analysis shows that PSD is substantially more efficient than standard SD. MINEDRAFT realizes the PSD through a novel batch-parallel design that maintains two batches of requests, overlapping drafting for one batch with verification for the other. Our experimental results show significant improvements of MINEDRAFT in both throughput (up to 75%) and end-to-end latency (up to 39%) over standard SD. Furthermore, we have implemented MINEDRAFT as a plugin for vLLM, demonstrating its practicality for production-ready inference systems. The code is publicly available in the MineDraft GitHub repository.

## 1. Introduction

Speculative decoding (SD) (Leviathan et al., 2023; Ryu & Kim, 2024; Xia et al., 2024) is a popular method used for accelerating large language model (LLM) inference. Instead of generating tokens sequentially with the target large model, SD uses a typically smaller and faster draft model to autoregressively draft (propose) a short sequence of tokens, which the target model then verifies in parallel. This parallel verification allows SD to achieve higher generation throughput than conventional autoregressive decoding when most drafted tokens are accepted. However, the effectiveness of SD is highly sensitive to the acceptance rate of drafted tokens, as drafting introduces additional computational overhead, frequent rejections can negate its benefits, and even make SD slower than standard autoregressive decoding.

Prior work has improved the verification success rate (VSR) of draft tokens through methods such as specialized drafting models (Cai et al., 2024; Li et al., 2024a), tree-based drafting (Wang et al., 2025), and active draft token selection (Wu et al., 2025). While these methods increase the overall VSR, they often lengthen the drafting stage due to increased model complexity (Cai et al., 2024; Li et al., 2024a) or sampling overhead (Wang et al., 2025; Wu et al., 2025). In standard speculative decoding, drafting and verification are executed *sequentially*, placing drafting directly on the critical path and fundamentally limiting achievable speedups. This observation motivates parallelizing the two phases: since the draft model is typically much smaller than the target model, its computation can be effectively overlapped with the target model's verification. By parallelizing the drafting and verification stages, a smaller draft model can generate draft tokens without blocking the verification process. When combined with an effective drafting mechanism, parallel speculative decoding hides drafting latency and yields improved overall inference performance.

Since the verification stage inherently depends on the outputs of the drafting stage, implementing parallelism in speculative decoding is nontrivial (Leviathan et al., 2023). Existing methods (Wang et al., 2024; Xiao et al., 2024; Timor et al., 2025) typically break this dependency at the cost of substantial additional resources, such as increased GPU or memory usage (Wang et al., 2024; Timor et al., 2025), or require a dedicated training stage for the draft model (Xiao et al., 2024). While Liu et al. (2025a) proposes a method that parallelizes drafting and verification for a single request using pre-verify and post-verify strategies, scaling this method to batched requests (multi-request) remains challenging. Moreover, its performance degrades when prerequisites for post-verification are not satisfied, a limitation that becomes especially severe in batched request settings.

---

[*]Equal contribution  [1]College of Computing and Data Science, Nanyang Technological University, Republic of Singapore  [2]Singapore-MIT Alliance for Research and Technology Centre  [3]Department of Computer Science, National University of Singapore, Republic of Singapore  [4]CSAIL, MIT, Cambridge, MA, USA. Correspondence to: Arun Verma <arun.verma@smart.mit.edu>.

*Proceedings of the $43^{rd}$ International Conference on Machine Learning*, Seoul, South Korea. PMLR 306, 2026. Copyright 2026 by the author(s).

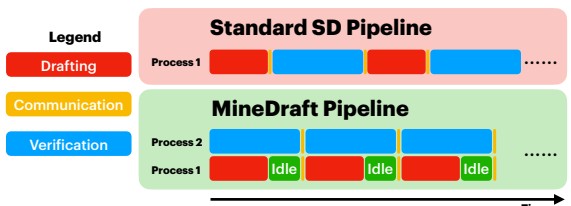

*Figure 1.* **MineDraft parallelizes drafting and verification:** a draft model generates tokens while the target model simultaneously verifies the previously generated draft tokens, thereby hiding drafting latency and improving overall inference throughput.

This paper presents MINEDRAFT, a framework for batch parallel speculative decoding (PSD) that systematically parallelizes the drafting and verification stages to hide drafting latency and thereby accelerate LLM inference. We provide a theoretical justification showing that PSD outperforms standard SD by efficiently utilizing additional computational resources through parallel drafting and verification. To efficiently parallelize drafting and verification, our framework uses a novel batch-parallel design that maintains two batches of requests, overlapping drafting for one batch with verification for the other. The name of our framework is inspired by the *Minecraft game engine*, which loads the next 'chunk' of blocks (the next set of draft tokens) into the game world while the player is still interacting with the current blocks (verifying the current drafted tokens). This 'background loading' is exactly how MINEDRAFT hides drafting latency.

Since MINEDRAFT is agnostic to the strategies for generating and selecting draft tokens, it can be combined with existing SD methods such as EAGLE* (Li et al., 2024a;b; 2025), PEARL (Liu et al., 2025a), and TETRIS (Wu et al., 2025), and is orthogonal to most LLM inference optimization techniques, including forward-pass modifications, model parallelism, and draft models. Our experimental results show that MINEDRAFT outperforms standard SD by achieving average throughput gains and reducing end-to-end latency at the cost of only 1 additional GPU, highlighting its significant practical impact. Furthermore, MINEDRAFT is implemented as a plugin for the production-ready inference library vLLM (Kwon et al., 2023). The specific contributions of this paper are summarized as follows:

- **Theoretical efficiency of PSD.** In Sec. 4, we first present a theoretical analysis of batch PSD, demonstrating that it achieves a 37.9% reduction in inference time compared to standard SD under mild assumptions about the quality and acceptance rate of the draft model.

- **A framework for batch PSD.** In Sec. 5, we propose a novel framework, MINEDRAFT, for batch PSD that maintains two batches of requests, overlapping drafting for one batch with verification for the other. To enable parallel execution, MINEDRAFT runs the draft model on a separate GPU and employs direct GPU-to-GPU communication to transfer draft tokens to the target model, which runs on GPUs separate from the draft model.

- **Experimental results.** In Sec. 6, we evaluate the performance of MINEDRAFT through comprehensive experiments across existing SD methods. Our results show that MINEDRAFT achieves average throughput gains of up to 75% and reduces end-to-end latency by up to 39% compared to standard SD, at the cost of one extra GPU.

- **MINEDRAFT as a vLLM plugin.** We develop a plug-and-play vLLM plugin that implements MINEDRAFT, encouraging further academic research and experimentation. The plugin also supports continuous batching (iteration-level scheduling) (Yu et al., 2022) and is fully compatible with PagedAttention (Kwon et al., 2023), highlighting its significant practical impact and ease of use.

## 2. Related Work

**Speculative decoding.** SD has been receiving increasing attention as an effective method to accelerate LLM inference by decomposing autoregressive decoding into a drafting phase and a verification phase (Leviathan et al., 2023; Ryu & Kim, 2024; Xia et al., 2024). Prior work has focused on improving SD performance in both single-request settings (Cai et al., 2024; Cheng et al., 2024; Zhang et al., 2024; Li et al., 2025; Oliaro et al., 2025) and batch-request settings (Li et al., 2026; Liu et al., 2025b; Wu et al., 2025), primarily by improving drafting efficiency. In contrast, MINEDRAFT explores an orthogonal direction by parallelizing the drafting and verification phases, allowing easy integration with existing methods to further improve SD performance.

**Parallel speculative decoding.** Recent work has explored reducing drafting overhead through parallelism. Parallel SD methods achieve faster decoding at the cost of substantial GPU resources, such as increased VRAM consumption (Wang et al., 2024; Timor et al., 2025). Another promising direction is parallel decoding with tree attention, which accelerates SD by drafting multiple tokens in parallel (Cai et al., 2024; Li et al., 2024a). In addition, prior work focuses on the single-request setting and modifies the SD process to generate variable-length drafts, thereby reducing synchronization overhead between the drafter and verifier (Liu et al., 2025a). Whereas MINEDRAFT adopts a new parallelism paradigm, *Batch Parallelism*, requiring only one additional GPU over standard SD, balancing resource usage with reduced overhead, and eliminating a dedicated training stage.

**Adaptive drafting.** Under many concurrent requests, standard SD can have suboptimal performance when using long speculative lengths, as the increased number of draft tokens places significant pressure on the verification stage. To mitigate this issue, adaptive drafting strategies generate variable-length drafts and selectively verify only the most promising tokens. EAGLE-2 (Li et al., 2024b) adopts this approach by expanding and reranking a context-aware draft tree. Other heuristics include selecting drafts with the high-

est likelihood of acceptance within a batch (Wu et al., 2025) and choosing among drafts generated by different speculative methods based on their historical acceptance rates (Hou et al., 2025). Additionally, AdaSpec (Huang et al., 2025) dynamically controls the execution of the draft model based on estimated draft quality and generation speed. Despite these advances, existing adaptive drafting methods yield limited performance gains: the number of draft tokens that can be generated remains constrained, as excessive drafting quickly makes verification overhead prohibitive.

## 3. Preliminaries

**Speculative decoding (SD).** We consider the standard SD setting, which involves a small draft model (the drafter) and a large target model (the verifier). At each SD step, the drafter first generates a sequence of draft tokens, after which the verifier evaluates them in parallel to decide whether to accept or reject them (i.e., verification), and then produces the final output. Standard SD does not allow parallelization between drafting and verification, since the verification inherently depends on the prior availability of the draft tokens.

**Parallel speculative decoding (PSD).** We consider a variant of PSD that systematically parallelizes the drafting and verification stages. By overlapping these stages, the draft model generates tokens for one batch of requests while the target model simultaneously verifies the draft tokens from another batch, thereby effectively hiding drafting latency and providing a significant speedup over standard SD.

**Notations.** We consider an LLM inference service provider with a fixed capacity $C$, representing the maximum number of tokens its computing resources can process in parallel at each decoding step. In practice, this capacity depends on the service provider's server configuration. At each decoding step, the server processes a batch of $m$ requests $r_1, r_2, \cdots, r_m$. Each request $r_i$ is associated with a partially generated sequence $S_{i,t_i} = (d_{i,1}, \ldots, d_{i,t_i})$, where $t_i$ represents the number of tokens generated so far for request $r_i$. We allow a variable draft window size for each request, i.e., $k_i$ for request $r_i$. At decoding step $s$, the draft model $\mathcal{S}$ drafts and selects a set $\mathcal{D}_s \coloneqq \{(i,j)|i \in \{1,\ldots,m\}, j \in \{t_i+1,\ldots,t_i+k_i\}\}$, such that the total number of selected tokens satisfies $|\mathcal{D}_s| = \sum_{i=1}^m k_i = C$.

**Average throughput.** Let $\mathbf{1}_{i,j}$ be an indicator variable that equals 1 if the token $(i,j) \in \mathcal{D}_s$ is verified, 0 otherwise. The average throughput is defined as the total number of verified tokens generated over $T$ steps, divided by the total time $\tau$ required to complete these steps:

$$\mathcal{G} \coloneqq \tau^{-1} \sum_{s=1}^T \left(\sum_{(i,j)\in\mathcal{D}_s} \mathbf{1}_{i,j} + m\right).$$

**End-to-end latency (E2EL).** The end-to-end latency is defined as the average time from request arrival to completion across a set of requests under the SD inference system.

## 4. Theoretical Efficiency of PSD

To highlight the fundamental differences between SD and PSD, we present an analysis under the constraint that the number of tokens sent for verification at each step is fixed.

**Drafting Pareto frontier.** Let $f : \mathbb{R}^+ \to [0, 1]$ denote the *Pareto frontier* of the draft model, characterizing the *maximum achievable verification success rate* given a drafting time budget. The function $f$ can be modified by changing the draft model or by applying speculating techniques such as Medusa (Cai et al., 2024), EAGLE (Li et al., 2024a), and TETRIS (Wu et al., 2025). Our goal is to compare the inference time of SD, denoted by $T_{\text{SD}}$, and that of PSD, denoted by $T_{\text{PSD}}$. For simplicity, we ignore the time required to draft the first batch of requests in the PSD analysis.

**Analysis.** We assume that the drafting time per step is a constant $t$. Let $R$ denote the total number of tokens to be generated across all requests. Under standard SD, the expected inference time incurred is

$$T_{\text{SD}} \coloneqq \min_t \mathbb{E}\left[\sum_{s=1}^T (t + V)\right],$$

where $T$ is a random variable representing the number of steps required to generate responses for all requests such that $\mathbb{E}[T] = R/f(t)$. The term $V$ denotes the verification time per step, which we assume to be a constant since the number of tokens submitted for verification is fixed. In practice, $f$ is typically an increasing function of $t$. Since $f(\cdot) \in [0, 1]$, it follows that $T_{\text{SD}} \to \infty$ as $t \to \infty$. On the other hand, under PSD, the drafting and verification times overlap, resulting in the expected inference time

$$T_{\text{PSD}} \coloneqq \min_t \mathbb{E}\left[\sum_{s=1}^T \max(V, t)\right] = \min_t \frac{R\max(V,t)}{f(t)}.$$

Under a mild assumption on $f$, we have the following result.

**Theorem 1.** *Let $f(t) = 1 - e^{-\alpha t}$ for some constant $\alpha \in \mathbb{R}^+$. For $\alpha V \approx 2.23$, we have $T_{\text{SD}} \gtrsim 1.61\, T_{\text{PSD}}$.*

This result shows that the guaranteed reduction in inference time is maximized at the operating point $\alpha V \approx 2.23$, where PSD achieves a $(1.61 - 1)/1.61 \approx 37.9\%$ reduction compared to standard SD. Since the larger values of $\alpha$ yield stronger diminishing returns in the verification success rate, the quantity $\alpha V$ can be interpreted as a proxy for the expected number of accepted tokens per verification step. From this perspective, any method with an improved Pareto frontier for $f$ results in more accepted tokens per step, thereby makes PSD more effective than standard SD. Finally, note that the theoretical upper bound on the improvement is $50\%$, corresponding to the idealized case in which drafting is perfectly hidden within the verification time and thus contributes no extra delay to the critical path.

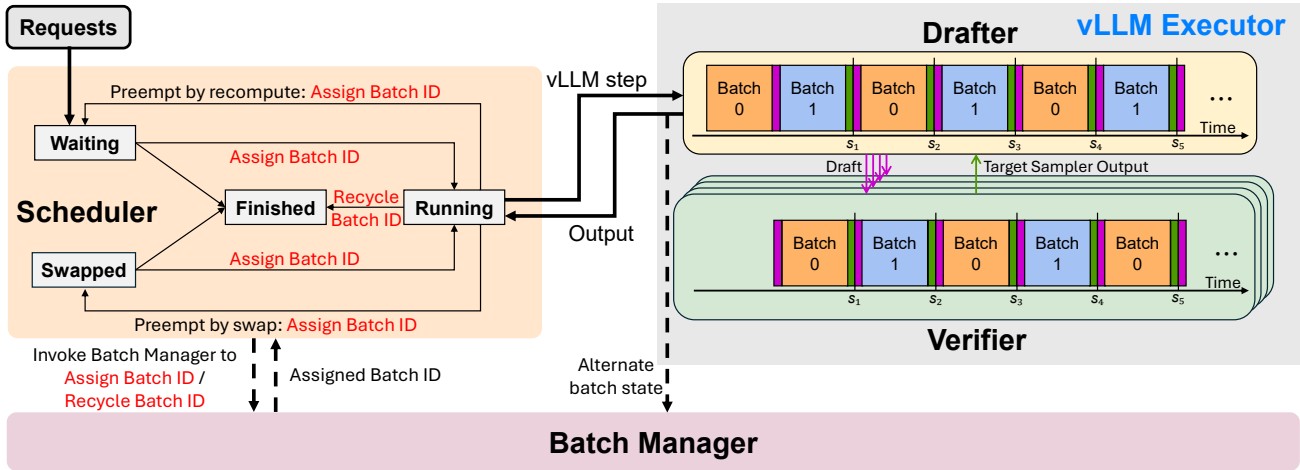

*Figure 2.* **Architecture overview of MINEDRAFT.** (Left) The **Scheduler** manages request life-cycles and batch IDs by coordinating with the **Batch Manager**, which maintains two batches to enable parallelism in MINEDRAFT. (Right) Parallel execution timeline of the **Drafter** and **Verifier** across speculative decoding (SD) steps. Magenta blocks/arrows denote broadcast of drafts from the Drafter to the Verifier, while dark green blocks/arrows denote point-to-point dispatch of target sampler outputs from the Verifier to the Drafter. Ticks indicate synchronization points between SD steps. During the initial SD step (before $s_1$), the Drafter sequentially drafts Batch 0, broadcasts drafts to the Verifier, then drafts Batch 1. At the same time, the Verifier immediately processes Batch 0 upon receipt and returns outputs to the Drafter. In subsequent SD steps, the two batches alternate roles between drafting and verification, enabling overlapped execution. Batch alternation is triggered when the Drafter returns outputs to the Scheduler at the end of each SD step.

## 5. MINEDRAFT: A Framework for Batch Parallel Speculative Decoding

We illustrate the overall architecture of MINEDRAFT in Fig. 2, which consists of a *Batch Manager*, a *Scheduler*, a *Drafter*, and a *Verifier*. Our parallelism follows a simple principle: at the beginning of each SD step, we regroup requests by filtering out those verified in the previous step, allowing the Verifier to continue operating in parallel with the Drafter. To fully utilize the Verifier, if it processes $m$ requests per step, we maintain $2m$ requests partitioned into two batches and alternate between them, sending $m$ requests to the Verifier at each SD step. We describe the role of each component below and provide a concrete example of a MINEDRAFT execution in Sec. C.2.

### 5.1. Batch Manager

**State variables and initialization.** Given a maximum of $m$ requests processed per step, the system supports up to $2m$ concurrent requests, which the Batch Manager organizes into two batches. Each running request is assigned a batch ID, either $0$ or $1$, indicating membership in Batch 0 or Batch 1, respectively. The imbalance between the two batches is tracked by a state variable `balance`, defined as the difference in their sizes: `balance` $=$ $|\text{Batch } 1| - |\text{Batch } 0|$. A positive `balance` indicates that Batch 1 contains more requests, in which case new requests are assigned to Batch 0 for re-balancing; a negative value indicates the opposite. At each SD step, the state variable `skip_batch` identifies the batch undergoing parallelized drafting, referred to as the *draft batch*. The remaining batch,

whose batch ID differs from `skip_batch`, is the *target batch*, whose draft tokens are verified concurrently in the same SD step. At initialization, `balance` is set to 0, indicating equal batch sizes, and `skip_batch` is set to 1, designating Batch 1 as the initial draft batch.

**Batch ID assignment.** In the first SD step, the `assign` operation assigns a batch ID to each new running request and serves as a load-balancing mechanism between the two batches. If `balance` $\geq 0$, indicating that Batch 1 has at least as many requests as Batch 0, the request is assigned to Batch 0 and `balance` is decremented. Otherwise, if `balance` $< 0$, the request is assigned to Batch 1 and `balance` is incremented. This assignment rule maintains a balanced workload between the two batches.

After the first SD step, assuming both batches have reached their capacity of $m$ requests, new running requests are assigned to the current `skip_batch` to ensure verification occurs only when draft tokens are available. This policy typically preserves balance between the two batches, since after the first SD step, a waiting request can be scheduled only when a request in the target batch finishes. However, exceptional terminations due to preemption or abort can disrupt this balance. Moreover, when chunked prefill is enabled, the Batch Manager may receive fewer than $2m$ requests from the Scheduler in the first SD step, resulting in batch sizes that fall below the capacity $m$. As subsequent requests are consistently assigned to the draft batch, this initial shortfall can lead to an irrecoverable workload imbalance between the two batches, degrading the expected performance gains of MINEDRAFT. We further discuss the implications of this

limitation and possible mitigation in Sec. A.

**Batch ID recycling.** The `recycle` operation is invoked when a request is finished, preempted, or aborted. It takes the batch ID $b$ of the terminated request as input, and updates `balance` to reflect the new distribution of requests. If the request belonged to Batch 0 ($b = 0$), `balance` is incremented; if it belonged to Batch 1 ($b = 1$), `balance` is decremented. This update is the inverse of the `assign` operation, preserving consistency of `balance`.

### 5.2. Drafter and Verifier

**Start-up of parallelism.** Batch Parallelism uses a *Drafter* and a *Verifier*. In the first SD step, the Drafter generates drafts for the target batch and sends them to the Verifier. While the Verifier processes the target batch, the Drafter concurrently proposes draft tokens for the draft batch. After verification, the Verifier returns the sampled tokens and probability distributions (target sampler output) to the Drafter.

**Batch state alternation.** At the end of each SD step, the Verifier sends the target sampler's output to the Drafter, which processes it and returns the output to the Scheduler. We refer to the moment when the Drafter returns output to the Scheduler as a ***sync point***. At each sync point, `skip_batch` is alternated with the other batch. After the first SD step, the Verifier verifies the target batch corresponding to the previous step's draft batch, while the Drafter concurrently proposes draft tokens for the new draft batch, which corresponds to the previous step's target batch.

**Fallback mechanism.** A corner case arises when one batch becomes empty as requests are finished and no new requests arrive. In this case, Batch Parallelism falls back to standard SD if draft tokens of the target batch are unavailable. This exposes a native limitation of MINEDRAFT: when the two batches cannot be replenished to maintain balance, it eventually leads to an empty batch and degrades performance. We discuss this limitation and a potential mitigation in Sec. A.

### 5.3. Scheduler

**KV block management.** In PagedAttention, a *KV block* stores the key–value vectors for a fixed number of tokens, defined by the *KV block size* ($B$). KV blocks associated with a request need not be contiguous in physical memory, allowing flexible memory management (Kwon et al., 2023). Before token generation, the Scheduler allocates the required KV blocks for each running request.

**Over-allocation issue in default scheduler.** We observe that during drafting, the Drafter accesses newly allocated KV blocks without populating them, whereas during verification, the Verifier computes and writes KV caches to the allocated blocks. The default vLLM scheduler assumes that all running requests are generated in each step and allocates

KV blocks for all of them. Under Batch Parallelism, this causes redundant allocations for requests in the draft batch, which are drafted but not verified, leaving allocated blocks accessed but unwritten. We patch the vLLM scheduler's KV block allocation logic to maintain compatibility with PagedAttention and avoid unnecessary memory allocation.

**Scheduler patch.** To implement this patch, we introduce a set, `has_deferred`, that tracks request IDs for which KV block allocation skipping has been deferred. The modified allocation logic proceeds as follows. If Batch 0 or Batch 1 is empty, the Scheduler allocates KV blocks for all running requests, except for the first occurrence of this condition. When both batches are non-empty, KV blocks are allocated for prefill requests to support the next prefill iteration. For requests in the decoding stage, KV blocks are allocated if the request ID is not in `has_deferred` or if the request belongs to the current draft batch; otherwise, allocation is skipped. After this decision, the request ID of each decoding request is added to `has_deferred`. This ensures that KV block allocation is not inadvertently skipped for requests in the target batch during the first SD step.

## 6. Experiments

We evaluate the effectiveness of MINEDRAFT using the performance metrics introduced in Sec. 3, namely average throughput (tokens/second or tokens/s) and end-to-end latency (milliseconds or ms). In addition to demonstrating the performance gains of MINEDRAFT, our empirical results show that MINEDRAFT can further improve speculative decoding by integrating existing drafting techniques.

**Model settings.** We consider seven target–draft model configurations, listed below. In all settings, the target model is served with tensor parallelism set to four, while a single GPU is dedicated to serving the draft model. Additional setup details are provided in Sec. C.1.

- **Settings 1–4** use Qwen3-32B (Yang et al., 2025) as the target model with different draft models: Qwen3-0.6B (Setting 1), Qwen3-1.7B (Setting 2), Qwen3-4B (Setting 3), Qwen3-8B (Setting 4). All are served on 5 L40S GPUs with $m = 16$.
- **Setting 5** pairs Llama-3.3-70B-Instruct-AWQ-INT4 (Ibanez, 2024) as the target model with Llama-3.1-8B-Instruct (Grattafiori et al., 2024) as the draft model, served on five H100 GPUs with $m = 64$.
- **Settings 6 and 7** use vicuna-33b-v1.3 and vicuna-13b-v1.3 (Zheng et al., 2023) as the target models, respectively, paired with EAGLE-Vicuna-33B-v1.3 and EAGLE-Vicuna-13B-v1.3 (Li et al., 2024a) as the draft model, respectively. These settings are served on five L40S GPUs with $m = 16$.

**Datasets.** Following the experimental settings of prior

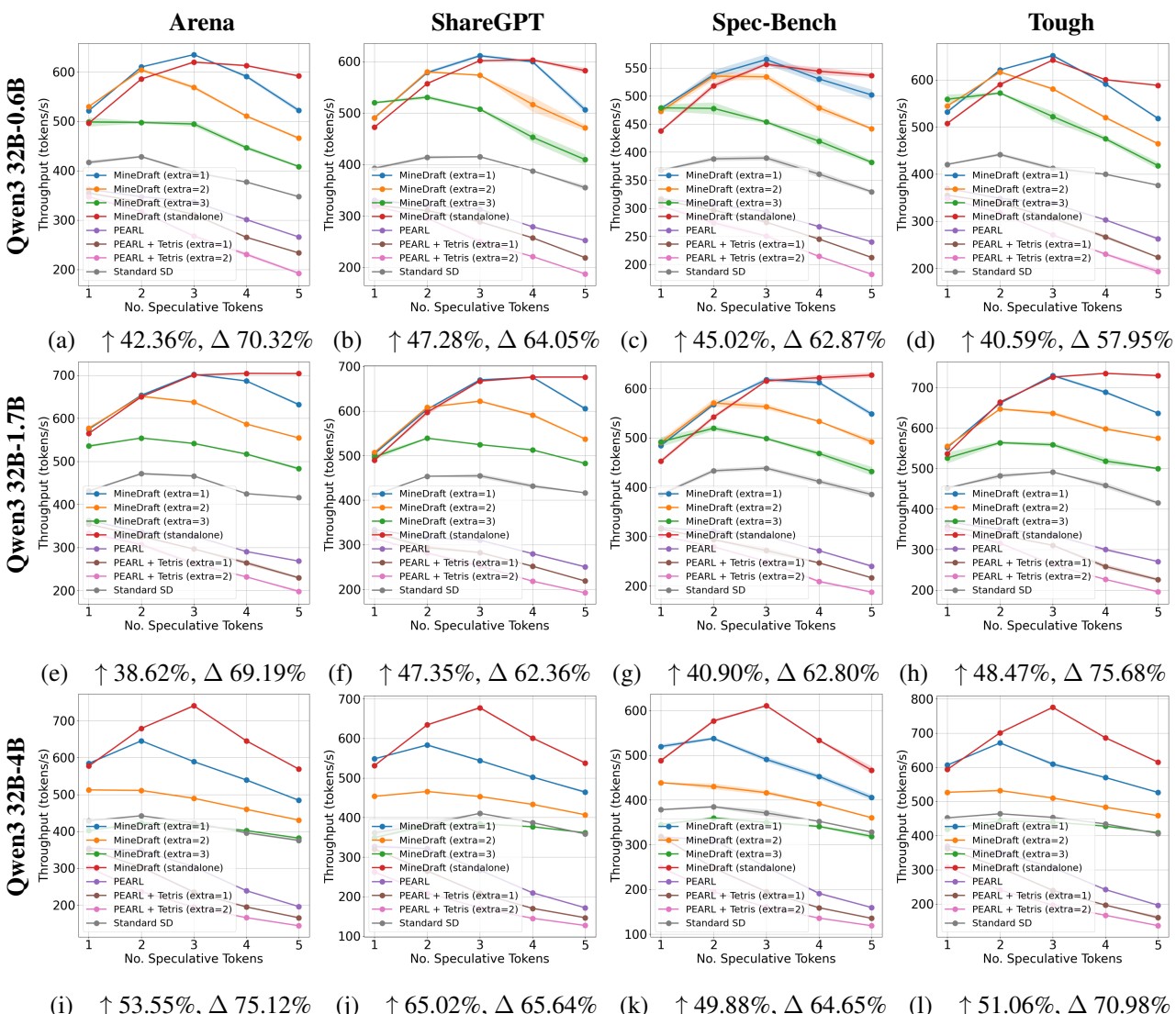

*Figure 3.* Throughput comparison against baseline methods across Settings 1–3. ↑ indicates the average improvement over the best baseline method. Δ indicates the maximum average gap between MINEDRAFT and standard SD. More details about ↑ and Δ are provided in Sec. C.3. MINEDRAFT consistently outperforms baselines, improving average throughput by up to 65.02% over the best-performing baseline and by up to 75.68% over standard SD.

work (Hou et al., 2025; Oliaro et al., 2025; Wu et al., 2025), we evaluate MINEDRAFT on ShareGPT (Anon, 2023) (Apache License 2.0), Arena (Zheng et al., 2023) (CC), LLM-Tough-Questions (Tough) (YAV-AI, 2024) (MIT License), and Spec-Bench (Xia et al., 2024) (Apache License 2.0), to robustly assess the applicability of MINE-DRAFT across a diverse range of scenarios.

### 6.1. Comparison against Baselines.

We measure the average throughput of MINEDRAFT and establish baselines, including PEARL (Liu et al., 2025a) and standard SD. As shown in Fig. 3, MINEDRAFT improves average throughput over the best-performing baseline by

38.62% to 65.02%, with maximum gains over standard SD ranging from 57.95% to 75.68%. These results show MINE-DRAFT consistently outperforms all baselines across all model settings and datasets. For all experimental results, we report the mean and standard deviation over three independent trials unless otherwise specified. We omit PEARL from subsequent experiments, as it exhibits substantially lower efficiency than MINEDRAFT in preceding evaluations.

**Issue with extreme $k$ value.** Notably, increasing the number of draft tokens per step is not always beneficial: excessive drafting can cause the draft stage to dominate verification time, thereby negating the benefit of hiding drafting latency via PSD. This effect is particularly significant for large draft models and is further analyzed in Sec. C.6.

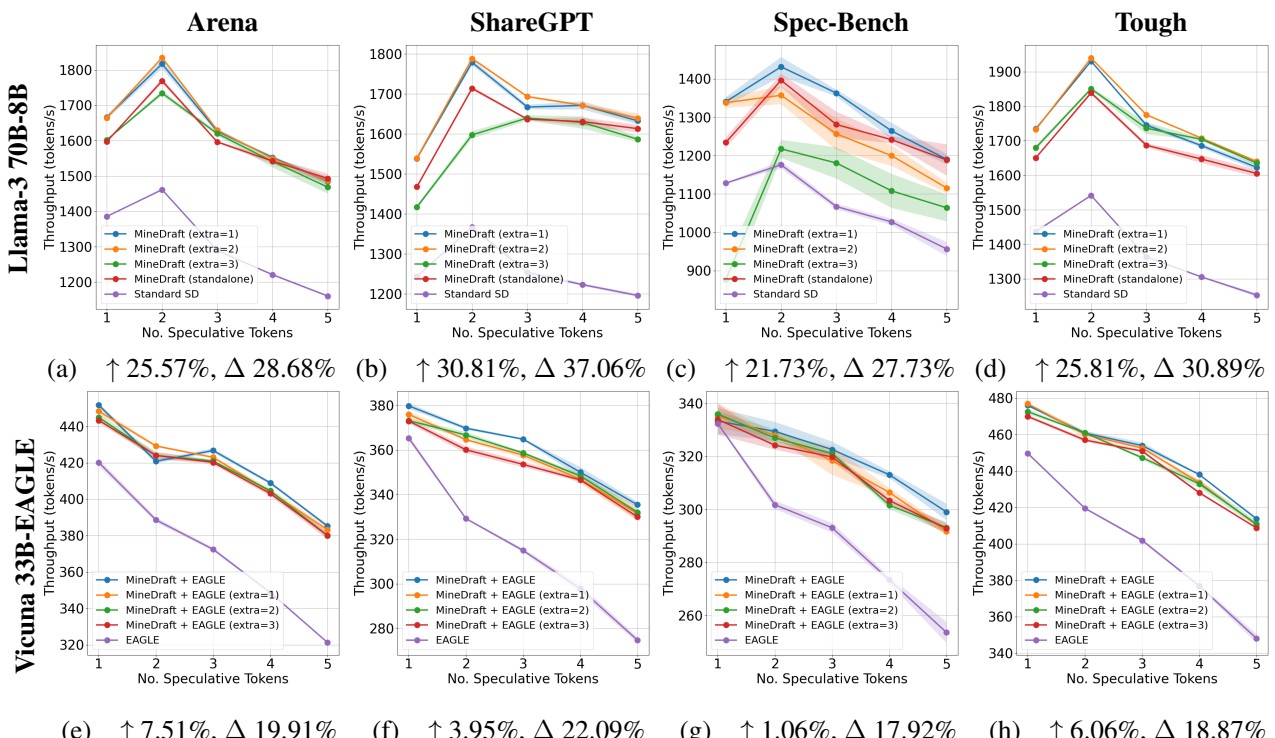

Figure 4. Throughput comparison across Settings 5 and 6. ↑ indicates the average improvement over the best baseline method. Δ indicates the maximum average gap between MINEDRAFT and standard SD or EAGLE. MINEDRAFT consistently outperforms standalone EAGLE and standard SD, achieving maximum average throughput gains of 37.06% and 22.09%, respectively.

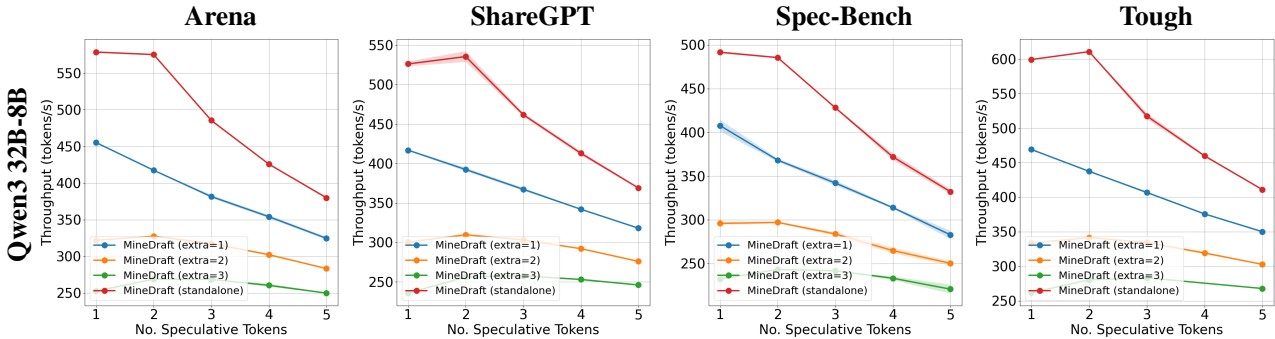

Figure 5. Throughput results on Setting 4. Standalone MINEDRAFT outperforms MINEDRAFT's integration with TETRIS when draft model is large. Standard SD experiments failed on this setting due to OOM.

## 6.2. Results with Different Drafting Strategies

We further evaluate MINEDRAFT integrated with other drafting strategies, including EAGLE and TETRIS, on the other model settings. As shown in Fig. 4, integrating MINEDRAFT with TETRIS and/or EAGLE consistently outperforms standalone EAGLE and standard SD across all datasets. In particular, MINEDRAFT improves average throughput by up to 30.81% and 7.51% over the best results of standard SD and EAGLE, respectively. The maximum average improvements over standard SD and EAGLE reach 37.06% and 22.09%, respectively.

**Memory contention in standard SD.** As shown in Fig. 5, standard SD can encounter memory limitations when de-

ploying larger draft models, such as Qwen3-8B, due to the coexistence of draft and target models within the same GPU memory, leading to VRAM contention. In contrast, the MINEDRAFT architecture places the draft model on a dedicated GPU, enabling parallel drafting without competing for memory with the target model. This design alleviates memory bottlenecks and promotes a more balanced utilization of available GPU resources. Similar memory constraints arise for standard SD when deploying large target models, such as Qwen3-235B-A22B-Instruct-2507-FP8 (Yang et al., 2025), as detailed in Sec. C.10.

**EAGLE's subpar performance.** We observe that EAGLE's performance degrades as the number of draft tokens

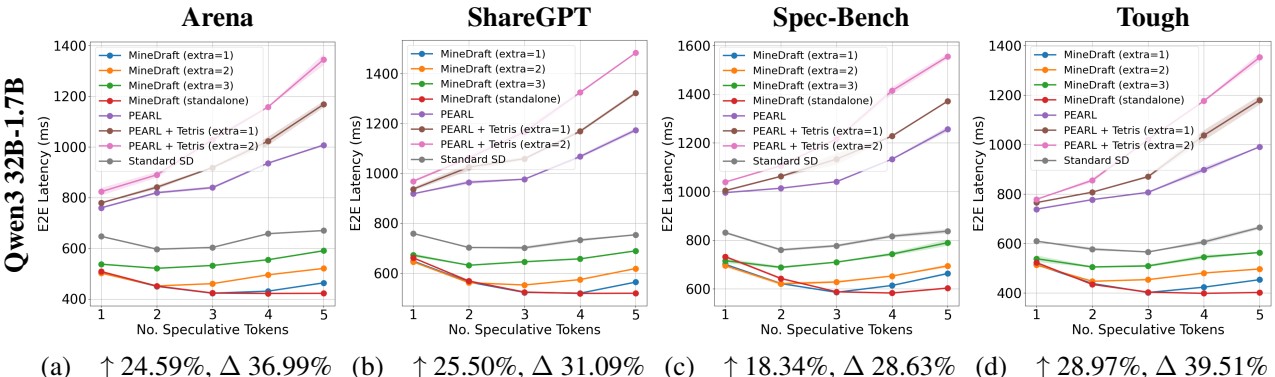

*Figure 6.* End-to-End Latency (ms) comparison against baseline methods in Setting 2. ↑ indicates the average improvement over the best baseline method. Δ indicates the maximum average gap between MINEDRAFT and standard SD. MINEDRAFT reduces latency by up to 28.97% compared to best-performing baseline and by up to 39.51% compared to standard SD.

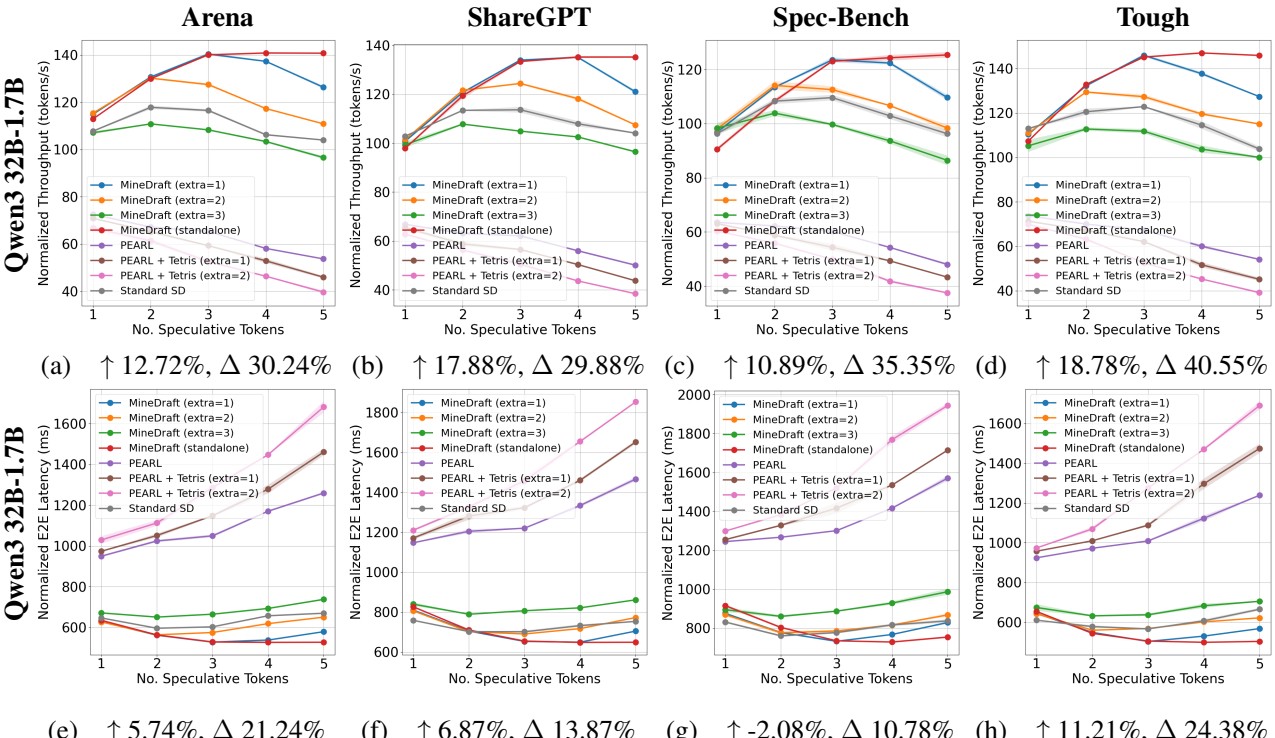

*Figure 7.* Normalized throughput and latency comparisons against baseline methods in Setting 2. ↑ indicates the average improvement over the best baseline method. Δ indicates the maximum average gap between MINEDRAFT and standard SD. MINEDRAFT improves normalized average throughput and latency by up to 40.55% and 24.38% over standard SD, respectively. Additional normalized results across other model settings are included in Sec. C.5.

per step increases, potentially due to the current vLLM implementation of EAGLE-based speculators, which is under investigation by the vLLM team (vLLM, 2024).

**MINEDRAFT as a batch PSD framework.** When integrated with TETRIS (denoted by "extra=" in the legends), MINEDRAFT outperforms its standalone variant under certain settings, highlighting its potential for further inference optimization when paired with effective drafting strategies. A more detailed breakdown of performance differences between integrations with drafting strategies and the stan-

dalone MINEDRAFT implementation is left for future work.

## 6.3. End-to-End Latency

We measure end-to-end latency of MINEDRAFT and baseline methods across drafting strategies, selected model settings, and datasets. Fig. 6 shows representative results for Setting 2 across all datasets, where MINEDRAFT demonstrates a 28.97% latency reduction relative to the best-performing baseline and a maximum reduction of 39.51% compared to standard SD. Complete results are in Sec. C.4.

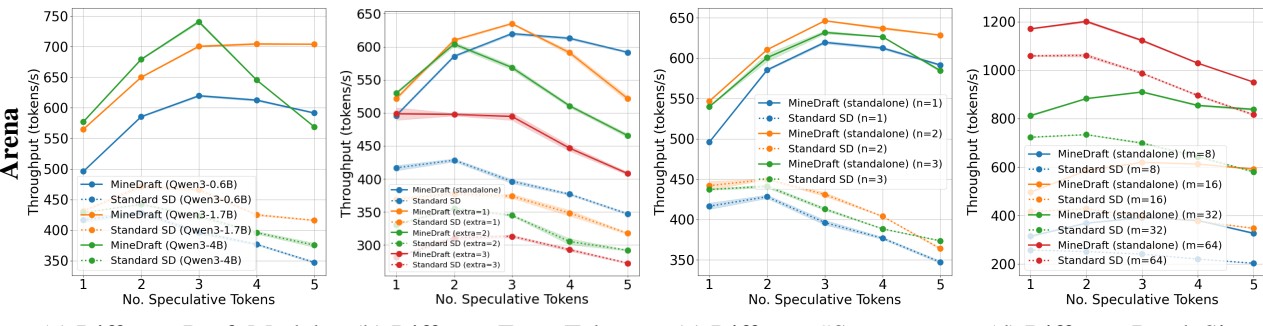

(a) Different Draft Models    (b) Different Extra Tokens    (c) Different #Sequences    (d) Different Batch Sizes

*Figure 8.* Different ablation studies using the Arena dataset. The comprehensive details and analysis of our ablation studies are included in Sections C.6 to C.8. Plot (a) indicates that the choice of draft model can affect MINEDRAFT's parallelism benefit. Plot (b) shows MINEDRAFT consistently improves average throughput relative to standard SD across different numbers of extra draft tokens generated for TETRIS. Plots (c) and (d) present how MINEDRAFT scales with larger $n$ (number of sequences per request) and $m$ (batch size) values.

### 6.4. Normalized Results w.r.t. GPU Count

To ensure a fair comparison between parallel methods (MINEDRAFT and PEARL, evaluated on 5 GPUs) and standard SD (evaluated on 4 GPUs), we normalize the metrics to a per-GPU basis, as specified in Sec. C.5. Fig. 7 illustrates that MINEDRAFT outperforms both standard SD and PEARL in normalized average throughput, achieving a 18.78% improvement over the best throughput achieved by baselines. Compared to standard SD, MINEDRAFT improves normalized average throughput by up to 40.55%. However, MINEDRAFT marginally underperforms standard SD in normalized latency when $k$ is small. Particularly, MINEDRAFT exhibits a 2.08% increase in normalized latency compared to the best-performing baseline at $k = 2$ on Spec-Bench. Additional comparisons of normalized metrics across Settings 1, 3, and 4, along with an analysis of this degradation in normalized latency, are provided in Sec. C.5.

### 6.5. Ablation Studies

This section presents representative ablation studies on the Arena dataset. Comprehensive results across all selected datasets are provided in Sections C.6 to C.8. Due to space constraints, analysis of verification success rate of integration with TETRIS and Nsight Systems profiling results is also provided in Sections C.9 and C.11.

**Draft model sizes.** We study the impact of draft model scale on system performance. Although MINEDRAFT consistently outperforms standard SD across draft model sizes, Fig. 8 shows that increasing the number of draft tokens per step does not monotonically improve average throughput. This degradation is most pronounced for Qwen3-4B: when more than three draft tokens are generated per step, throughput drops below that achieved with only two draft tokens. A detailed analysis of this behavior is provided in Sec. C.6.

**Extra tokens.** We evaluate the average throughput of MINEDRAFT and standard SD across varying numbers of extra draft tokens ($e$) for TETRIS. Fig. 8 presents results for Setting 1 on the Arena dataset, where MINEDRAFT attains its maximum throughput at $k = 3$ and $e = 1$. As $e$ increases, MINEDRAFT integrated with TETRIS outperforms standalone MINEDRAFT for $e \leq 2$, but experiences a degradation in throughput for $e \geq 3$. Complete results and a detailed analysis of this trend are provided in Sec. C.7.

**Scaling behavior with $n$ and $m$.** We evaluate throughput scaling with respect to the number of sequences per request ($n$) and batch size ($m$). The final two plots in Fig. 8 show that, with $m$ held constant, increasing $n$ yields only marginal throughput gains, whereas increasing $m$ consistently improves average throughput. Moreover, when the number of draft tokens per step exceeds 4, the average throughput of MINEDRAFT approaches that of standard SD with doubled batch size, suggesting that MINEDRAFT can complement adaptive drafting strategies, which are often limited by draft length, to enable additional performance gains. More experiments are provided in Sec. C.8.

## 7. Conclusion

This paper introduces MINEDRAFT, a novel framework that realizes the potential of PSD. MINEDRAFT exploits the latency savings allowed by *Batch Parallelism*, which overlaps drafting and verification in SD by maintaining a target batch and a draft batch. Our theoretical analysis characterizes the expected minimum end-to-end latency gains achievable with PSD. Extensive experiments across diverse model families, model sizes, hyperparameter settings, and datasets further validate the performance advantages of MINEDRAFT in practice. As future work, we will explore extending support to the vLLM v1 engine and the chunked prefill feature (Agrawal et al., 2023). We also plan to study whether the current requirement for an additional GPU in Batch Parallelism can be eliminated by leveraging the weight-padding method described in Appendix F of (Liu et al., 2025a).

## Acknowledgements

This research is supported by the National Research Foundation (NRF), Prime Minister's Office, Singapore under its Campus for Research Excellence and Technological Enterprise (CREATE) programme. The Mens, Manus, and Machina (M3S) is an interdisciplinary research group (IRG) of the Singapore MIT Alliance for Research and Technology (SMART) centre.

## Impact Statement

This paper proposes MINEDRAFT, a framework for batch parallel speculative decoding that improves throughput and reduces latency for LLM inference. By accelerating end-to-end generation, MINEDRAFT can broaden access to capable AI systems and improve the efficiency of shared inference infrastructure. Speculative decoding incurs additional computation relative to standard autoregressive decoding because it executes both a draft and a target model. Consequently, the overall impact on energy consumption depends on the deployment setting: improvements in serving throughput and hardware utilization may offset the increased FLOPs. However, such gains are not universal across workloads or system configurations. As with other advances in efficient text generation, reduced inference latency may lower the cost of producing content at scale, including in adversarial contexts such as spam or misinformation campaigns. We believe, however, that the benefits of faster and more accessible inference outweigh these risks. Mitigating harmful downstream use is better addressed through application-level safeguards, monitoring, and policy interventions than by limiting research on inference efficiency.

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

## A. Limitations

**Irrecoverable from biased batch workload.** The workload balance between batches can become irrecoverably disrupted under certain conditions. When a request is terminated due to user abort or preemption, MINEDRAFT may fail to recover from the resulting imbalance because new replacement running requests are consistently assigned to the draft batch. While this policy ensures that draft tokens are available for verification, it prevents the load-balancing mechanism (the `assign` operation) from correcting the induced batch imbalance. Furthermore, when chunked prefill is enabled, the Scheduler may initialize fewer than $2m$ running requests in the first SD step. In this case, MINEDRAFT must continue to apply the load-balancing `assign` operation for all subsequent batch ID assignments, rather than switching to the simplified post–first-step policy. Otherwise, an irrecoverable workload imbalance between the two batches can arise. However, continuously applying the `assign` operation introduces a minor overhead. If a new running request is assigned to the target batch, the Verifier can sample only a single token because there are no draft tokens, resulting in one less efficient "idle" SD step for that request. Since this cost is incurred at most once per request, it is deemed an acceptable trade-off for the long-term performance benefits of maintaining batch balance. Accordingly, we plan to adopt this revised batch ID assignment strategy in a future version of MINEDRAFT that supports the vLLM v1 engine and chunked prefill.

**Batch exhaustion and tail effect.** The Batch Parallelism scheme, as specified in Sec. 5, includes a fallback mechanism: when one batch becomes empty and the other batch contains no verifiable drafts, the system reverts to standard SD, which inevitably curtails the duration of parallel execution. Although Batch Parallelism successfully parallelizes the substantial majority of SD steps and yields significant performance gains per our empirical results, the tail effect circumscribes its maximum potential improvement. A possible approach to offset this limitation involves adopting a modified PEARL parallelism. When one batch becomes empty, the system instructs the Drafter to generate draft tokens for the remaining batch, including those undergoing verification, without confirming their success. In the next step, instead of verifying only previously successful requests as in standard PEARL, the system re-drafts requests that failed verification before submitting the batch to the Verifier. This re-drafting mechanism for failed verification attempts is intended for partially overlapping drafting time, optimizing performance to the greatest extent possible.

**Adaptive to varying workload.** Although MINEDRAFT supports continuous batching that automatically handles varying workloads, it is assumed that the number of requests in the system seldom drops below the nominal capacity of $2m$. If the actual total batch size of draft and target batches is frequently less than $2m$, MINEDRAFT may suffer from batch imbalance or even fall back to sequential SD, thereby diluting the parallelism effect. On the other hand, if the user-configured batch size $m$ is too large, it is desirable to automatically turn off speculative decoding or adaptively adjust $m$ to prevent the compute-bound scenarios (Liu et al., 2025b; Sadhukhan et al., 2025). It is left as a future direction to design and integrate an adaptive scheduling mechanism that accounts for resource constraints, selected drafter-verifier pair, and active workload.

**Request-dependent draft token selection.** In practice, draft acceptance rates are varied and correlated with the "difficulty" of different requests, and selecting draft tokens irrespective of request difficulty can be insufficient to attain the full potential of speculative decoding and MINEDRAFT. Existing literature has proposed a proxy measure to estimate the request difficulty (Agrawal et al., 2024), which can be employed by future work to develop a request-dependent draft selection algorithm that dynamically adjusts the number of draft tokens based on the predicted difficulty of each incoming request.

## B. Leftover Proof

We first derive an upper bound on $T_{\text{SD}}$ that does not involve the expectation operator. Under a mild assumption on the form of $f$, we have the following result.

**Lemma 1.** *Let $f(t) = 1 - e^{-\alpha t}$ for some constant $\alpha \in \mathbb{R}^+$. We have that*

$$T_{\text{SD}} > \frac{\frac{1}{\alpha}\ln(\alpha V + 1) + V}{1 - \frac{1}{2(\alpha V + 1)}} \times R \, .$$

*Proof.* First, the form of $f(x)$ is monotone increasing in the domain of $\alpha$. Hence, the function $g(t) = \frac{t+V}{f(t)}$ has a global minimum, taken at its turning point (critical point theorem). In the following, we derive this turning point $g'(t) = 0$. Taking the derivative with respect to $t$ and substituting the formula of $f(t)$, we have

$$g'(t) = 0$$

$$\Longleftrightarrow (t + V)f'(t) = f(t)$$
$$\Longleftrightarrow \alpha(t + V)e^{-\alpha t} = 1 - e^{-\alpha t} .$$

Rearrange the last equation:

$$(\alpha(t + V) + 1)e^{-\alpha t} = 1 .$$

We use the Lambert $W$ function to solve this equation. Specifically, the Lambert $W$ function is defined as the inverse function of $f(w) = we^w$, i.e.,

$$z = we^w \iff w = W(z) .$$

With this, let $u := \alpha(t + V) + 1$. Then, the equation above can be rewritten as $-ue^{-u} = -e^{-(\alpha V + 1)}$. So,

$$u = -W(-e^{-(\alpha V + 1)}) ,$$

and since $t = \frac{u}{\alpha} - (V + \frac{1}{\alpha})$, we have

$$t^* = -\frac{1}{\alpha}W(-e^{-(\alpha V + 1)}) - (V + \frac{1}{\alpha}) .$$

For $z \in (-e^{-1}, 0)$, there are two real branches: $W_0(z) \in (-1, 0)$ and $W_{-1}(z) \in (-\infty, -1]$. Here $z = -e^{-(\alpha V + 1)} \in (-e^{-1}, 0)$ and $u = \alpha(t + V) + 1 \geq 1$. If we took $W_0$, we would have $u = -W_0(z) \in (0, 1)$, a contradiction. Therefore, we take $W_{-1}$. Let $\mu := \alpha V + 1$. The expansion for the $-1$ branch as $\mu \to \infty$ is

$$W_{-1}(z) = \ln(-z) - \ln(-\ln(-z)) + \frac{\ln(-\ln(-z))}{\ln(-z)} + \cdots .$$

Substitute $z = -e^{-\mu}$, we have

$$W_{-1}(-e^{-\mu}) = -\mu - \ln\mu - \frac{\ln\mu}{\mu} + \mathcal{O}\left(\frac{(\ln\mu)^2}{\mu^2}\right) .$$

Plugging this equation into the equation of $t^*$, we have

$$t^* = \frac{1}{\alpha}\left[\ln\mu + \frac{\ln\mu}{\mu} + \mathcal{O}\left(\frac{(\ln\mu)^2}{\mu^2}\right)\right]$$
$$= \frac{1}{\alpha}\ln(\alpha V + 1) + \frac{1}{\alpha}\frac{\ln(\alpha V + 1)}{\alpha V + 1} + \mathcal{O}\left(\frac{(\ln(\alpha V))^2}{\alpha(\alpha V)^2}\right) .$$

With this, we now bound the range of $t^*$. First, recall that $(\alpha(t + V) + 1)e^{-\alpha t} = 1$. We rearrange the equation of $t$ into

$$\alpha t = \ln(\alpha(t + V) + 1) .$$

Let $y := \alpha t$ and define $g(y) := \ln(\alpha V + 1 + y) - y$. Then, $g'(y) = \frac{1}{\alpha V + 1 + y} - 1 < 0$, so $g$ is strictly decreasing and has a unique zero $y^*$. Define $y_0 := \ln(\alpha V + 1)$ and $y_1 := \ln(2(\alpha V + 1))$. We now show that $y_0 < y^* < y_1$.

At $y_0$, we have $g(y_0) = \ln(1 + \frac{\ln(\alpha V + 1)}{\alpha V + 1}) > 0$. This implies $y^* > y_0$.

At $y_1$, $g(y_1) = \ln(1/2 + \frac{\ln(2(\alpha V + 1))}{2(\alpha V + 1)})$. For $x \geq 1$, $\ln(2x) \leq x$. Hence, $1/2 + \ln(2x)/(2x) < 1$. Therefore, $g(y_1) < 0$ for $\alpha V > 0$. This implies $y^* < y_1$. Therefore, substitute $t = y/\alpha$, we have

$$\frac{1}{\alpha}\ln(\alpha V + 1) < t^* < \frac{1}{\alpha}\ln(2(\alpha V + 1)) .$$

As such, we have a bound for $T_{\text{SD}}$ as

$$T_{\text{SD}} = \frac{t^* + V}{f(t^*)} \times R > \frac{\frac{1}{\alpha}\ln(\alpha V + 1) + V}{f(\frac{1}{\alpha}\ln(2(\alpha V + 1)))} \times R = \frac{\frac{1}{\alpha}\ln(\alpha V + 1) + V}{1 - \frac{1}{2(\alpha V + 1)}} \times R . \qquad \square$$

On the other hand, under PSD, the expected time for verification is

$$T_{\text{PSD}} := \min_t \mathbb{E}\left[\sum_{s=1}^{T} \max(V, t)\right]$$
$$= \min_t \mathbb{E}\left[T\right] \max(V, t)$$
$$= \min_t \frac{R \max(V, t)}{f(t)} \ .$$

While it is obvious from the two definitions that $T_{\text{PSD}} < T_{\text{SD}}$, we prove a stronger result:

**Lemma 2.** *Similarly, let $f(t) = 1 - e^{-\alpha t}$ for some constant $\alpha \in \mathbb{R}^+$. We have that*

$$T_{\text{PSD}} = \frac{RV}{1 - e^{-\alpha V}} \ .$$

*Proof.* We minimize $h(t) := \frac{\max(V, t)}{f(t)}$ over $t > 0$. For $t < V$, we have $\max(V, t) = V$, so $h(t) = V/f(t)$, which is strictly decreasing since $f$ is strictly increasing. For $t \geq V$, we have $\max(V, t) = t$, so $h(t) = t/f(t)$. Computing the derivative, $h'(t) > 0$ if and only if $f(t) > tf'(t)$, i.e., $(1 - e^{-\alpha t}) > \alpha t e^{-\alpha t}$, which after multiplying through by $e^{\alpha t}$ is equivalent to $e^{\alpha t} - 1 > \alpha t$. This holds for all $t > 0$ by the standard inequality $e^x > 1 + x$ applied with $x = \alpha t > 0$, so $h$ is strictly increasing on $[V, \infty)$. Since $h$ is strictly decreasing on $(0, V)$ and strictly increasing on $[V, \infty)$, the unique global minimum is attained at $t^* = V$.

$$T_{\text{PSD}} = \min_t \frac{R \max(V, t)}{f(t)} = \frac{RV}{1 - e^{-\alpha V}} \ . \qquad \square$$

**Theorem 1.** *Let $f(t) = 1 - e^{-\alpha t}$ for some constant $\alpha \in \mathbb{R}^+$. For $\alpha V \approx 2.23$, we have $T_{\text{SD}} \gtrsim 1.61 \, T_{\text{PSD}}$.*

*Proof.* Using Lemma 1 and Lemma 2,

$$\frac{T_{\text{SD}}}{T_{\text{PSD}}} = \frac{\frac{1}{\alpha}\ln(\alpha V + 1) + V}{V} \frac{1 - e^{-\alpha V}}{1 - \frac{1}{2(\alpha V + 1)}} \ .$$

Substituting $\alpha V = 2.23$ gives $\frac{\frac{1}{\alpha}\ln(\alpha V + 1) + V}{V} \frac{1 - e^{-\alpha V}}{1 - \frac{1}{2(\alpha V + 1)}} \gtrsim 1.61111$, hence $T_{\text{SD}} \gtrsim 1.61 \, T_{\text{PSD}}$. $\qquad \square$

## C. Additional Experiment Results

### C.1. Experiment Setups

Unless otherwise specified, we use PyTorch 2.7.0 (PyTorch, 2025), HuggingFace Transformers 4.53.3 (Hugging Face, 2025), and vLLM 0.9.2 (vLLM, 2025), with temperature set to 1.0, one sequence per request ($n = 1$), and 256 output tokens per sequence (ignoring the end-of-sequence token). Target models are served with a tensor parallelism (TP) size of 4, and draft models with TP set to 1.

### C.2. Example Lifespan Illustration

**Timeline illustration.** Fig. 9 illustrates a Batch Parallelism lifespan with $m = 4$ and 12 total requests. In the first SD step, Batch 0 serves as the target batch and Batch 1 as the draft batch. The target batch is drafted and then verified in SD step 1. While the target batch is being verified in SD step 1, the draft batch is drafted in parallel. In subsequent SD steps, the two batches alternate states as target and draft batches, enabling concurrent drafting and verification. This alternating pattern continues until the target batch of an SD step lacks draft tokens (e.g., SD step 10 in Fig. 9), at which point the fallback to standard SD is triggered.

**KV block allocation illustration.** Fig. 10 depicts the same lifespan from the Scheduler's perspective, illustrating the patched KV block allocation logic. In the first SD step, KV blocks are allocated for all requests in both the target and draft batches. From SD step 2 through SD step 8, the Scheduler allocates KV blocks only for the draft batch, skipping

allocation for the target batch. At SD step 9, one batch becomes empty; since this is the first such occurrence, the patched logic continues to skip allocation. From SD step 10, the Scheduler resumes allocating KV blocks to all running requests at each step until all requests are completed.

**Drafting**

| Step1 | | Step2 | Step3 | Step4 | Step5 | Step6 | Step7 | Step8 | Step9 | Step10 | Step11 |
|---|---|---|---|---|---|---|---|---|---|---|---|
| Req1 | Req2 | Req1 | Req2 | Req1 | Req2 | Req1 | Req4 | Req10 | | Req10 | Req12 |
| Req3 | Req4 | Req3 | Req4 | Req3 | Req4 | Req10 | Req6 | Req11 | | Req11 | |
| Req5 | Req6 | Req5 | Req6 | Req5 | Req6 | Req11 | Req9 | Req12 | | Req12 | |
| Req7 | Req8 | Req7 | Req8 | Req7 | Req9 | Req12 | | | | | |

**Verifying**

| Step1 | | Step2 | Step3 | Step4 | Step5 | Step6 | Step7 | Step8 | Step9 | Step10 | Step11 |
|---|---|---|---|---|---|---|---|---|---|---|---|
| | | Req1 Req2 | Req1 | Req2 | Req1 | Req2 | Req1 | Req4 | Req10 | Req10 | Req12 |
| | | Req3 Req4 | Req3 | Req4 | Req3 | Req4 | Req10 | Req6 | Req11 | Req11 | |
| | | Req5 Req6 | Req5 | Req6 | Req5 | Req6 | Req11 | Req9 | Req12 | Req12 | |
| | | Req7 Req8 | Req7 | Req8 | Req7 | Req9 | Req12 | | | | |

Time →

*Figure 9.* **Timeline of an example Batch Parallelism lifespan.** Requests with an orange background belong to Batch 0, those with a blue background belong to Batch 1, and those with a red background have completed execution.

| Batch ID | Req ID | Step1 | | Step2 | Step3 | Step4 | Step5 | Step6 | Step7 | Step8 | Step9 | Step10 | | Step11 | |
|---|---|---|---|---|---|---|---|---|---|---|---|---|---|---|---|
| 0 | Req1 | Alloc | Fill | Alloc | Fill | Alloc | Fill | Alloc | FIN | | | | | | |
| 0 | Req3 | Alloc | Fill | Alloc | Fill | Alloc | FIN | | | | | | | | |
| 0 | Req5 | Alloc | Fill | Alloc | Fill | Alloc | FIN | | | | | | | | |
| 0 | Req7 | Alloc | Fill | Alloc | Fill | Alloc | FIN | | | | | | | | |
| 1 | Req2 | Alloc | | | Fill | Alloc | Fill | Alloc | FIN | | | | | | |
| 1 | Req4 | Alloc | | | Fill | Alloc | Fill | Alloc | Fill | Alloc | FIN | | | | |
| 1 | Req6 | Alloc | | | Fill | Alloc | Fill | Alloc | Fill | Alloc | FIN | | | | |
| 1 | Req8 | Alloc | | | Fill | Alloc | FIN | | | | | | | | |
| 1 | Req9 | | | | | | Alloc | Fill | Alloc | FIN | | | | | |
| 0 | Req10 | | | | | | | | Alloc | Fill | Alloc | Fill | Alloc | FIN | |
| 0 | Req11 | | | | | | | | Alloc | Fill | Alloc | Fill | Alloc | FIN | |
| 0 | Req12 | | | | | | | | Alloc | Fill | Alloc | Fill | Alloc | Fill | Alloc | FIN |

Time →

*Figure 10.* **Scheduler perspective of the example Batch Parallelism lifespan.** At each time point, `Alloc` indicates that the Scheduler allocates KV blocks for a request, `Fill` indicates that the allocated blocks are populated with updated KV vectors, and `FIN` indicates that the request has completed.

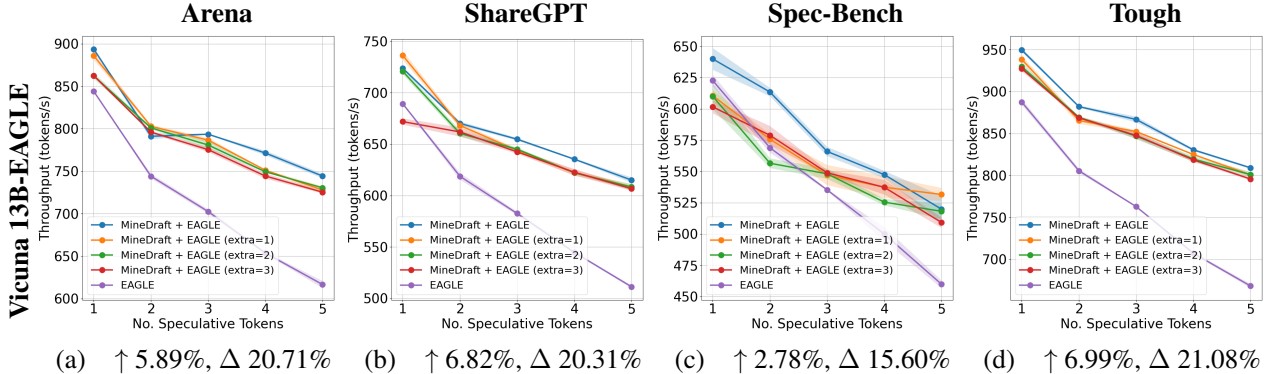

| Arena | ShareGPT | Spec-Bench | Tough |
|---|---|---|---|
| (a) ↑ 5.89%, Δ 20.71% | (b) ↑ 6.82%, Δ 20.31% | (c) ↑ 2.78%, Δ 15.60% | (d) ↑ 6.99%, Δ 21.08% |

*Figure 11.* Throughput comparison on Setting 7. ↑ indicates the average improvement over the best baseline method. Δ indicates the maximum average gap between MINEDRAFT and EAGLE. MINEDRAFT consistently outperforms EAGLE, improving average throughput by up to 6.99% over the best case of EAGLE and by up to 21.08% across all tested $k$.

### C.3. Performance Improvement Metrics.

To quantify the performance improvements of MINEDRAFT, we compute two percentage metrics relative to baseline methods. Let $p$ denote a performance metric, and let $B_p(k)$ denote the best value of $p$ achieved by baseline methods at $k$, where $k$ is the number of draft tokens per step. Similarly, let $M_p(k)$ denote the best value of $p$ achieved by MINEDRAFT variants (including the standalone variant and its integration with TETRIS) at $k$.

**Average improvement over the best baseline.** First, for each performance metric $p$, the optimal baseline draft token count $k_p^*$ is selected as follows:

- If $p$ is **maximized** (higher is better): $k_p^* = \arg\max_k B_p(k)$.

- If $p$ is **minimized** (lower is better): $k_p^* = \arg\min_k B_p(k)$.

Let $B_p^* = B_p(k_p^*)$ be the corresponding baseline performance metric value. Next, among all MINEDRAFT variants, select the best MINEDRAFT performance metric value at $k_p^*$ as $M_p^* = M_p(k_p^*)$. The average improvement of $p$ of MINEDRAFT over the best baseline method $\uparrow_p$ (in percent) is calculated as follows:

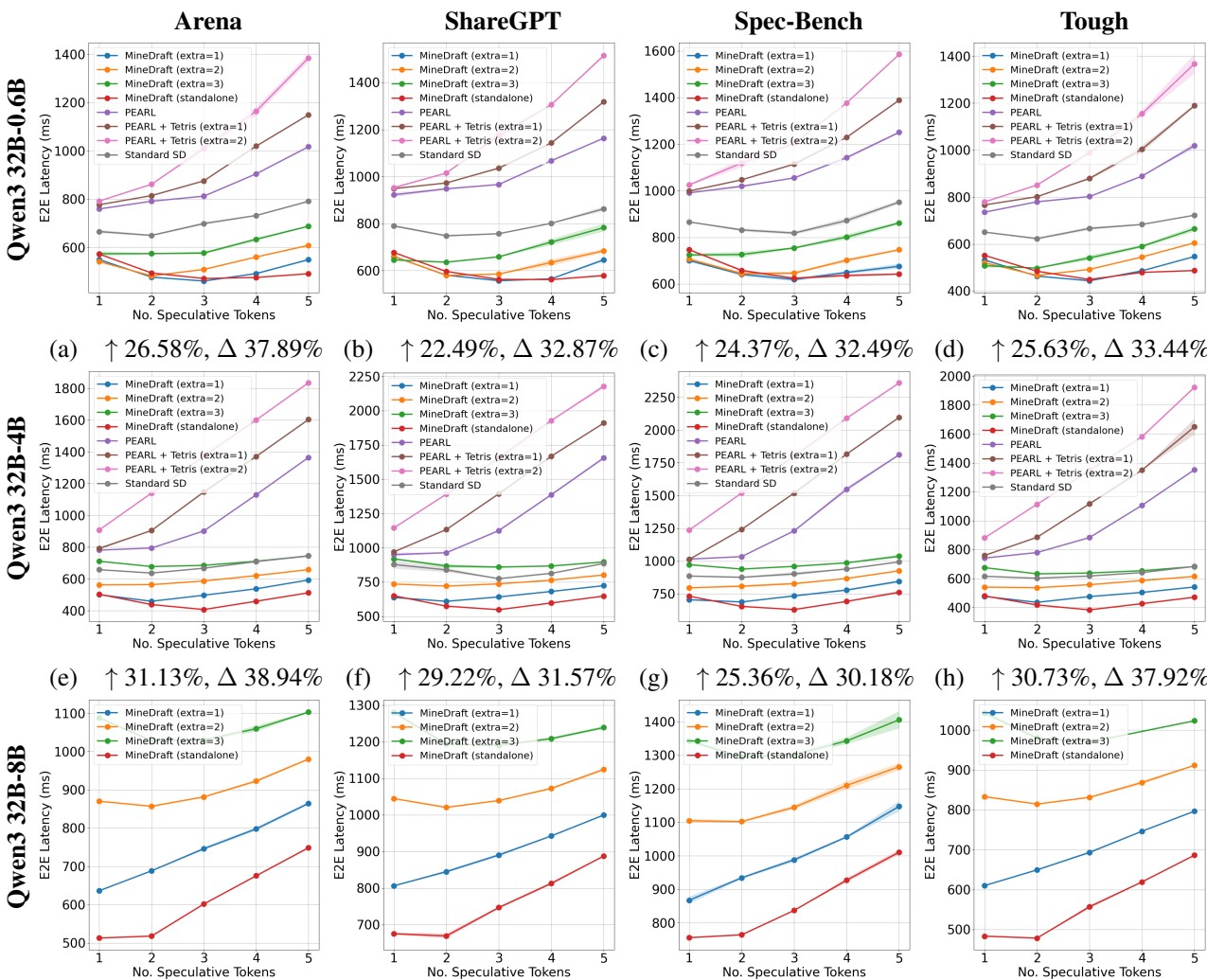

*Figure 12.* End-to-end latency comparison against baseline methods across Settings 1, 3, and 4. ↑ indicates the average improvement over the best baseline method. Δ indicates the maximum average gap between MINEDRAFT and standard SD. MINEDRAFT consistently outperforms baselines, achieving a reduction in latency of up to 31.13% compared to the best-performing baseline and up to 38.94% compared to standard SD. Standard SD fails in Setting 4 due to OOM.

- If $p$ is **maximized**: $\uparrow_p = \frac{M_p^* - B_p^*}{B_p^*} \times 100$.

- If $p$ is **minimized**: $\uparrow_p = \frac{B_p^* - M_p^*}{B_p^*} \times 100$.

**Maximum average improvement.** Define $S_p(k)$ as the value of $p$ of standard SD or EAGLE* (Li et al., 2024a;b; 2025), where applicable, at $k$. For a given target metric $p$, the average percentage improvement of MINEDRAFT over standard SD at $k$ ($\Delta_p(k)$) is calculated as:

- If $p$ is **maximized**: $\Delta_p(k) = \frac{M_p(k) - S_p(k)}{S_p(k)} \times 100$.

- If $p$ is **minimized**: $\Delta_p(k) = \frac{S_p(k) - M_p(k)}{S_p(k)} \times 100$.

The maximum average percentage improvement of $p$ of MINEDRAFT over standard SD or EAGLE ($\Delta_p$) is the maximum of $\Delta_p(k)$ over all $k$ values: $\Delta_p = \max_k \Delta_p(k)$.

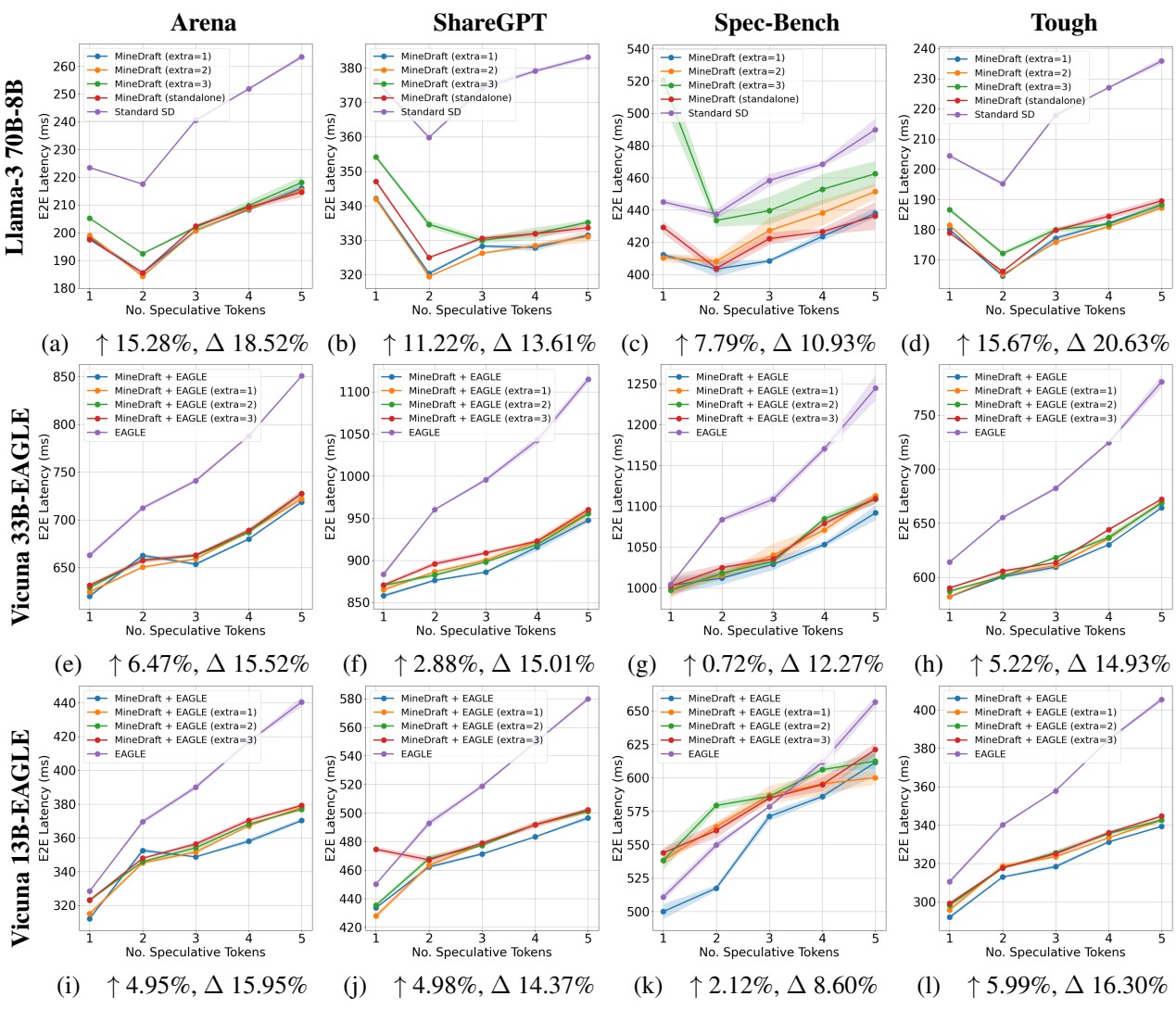

*Figure 13.* End-to-end latency comparison across Settings 5–7. ↑ indicates the average improvement over the best baseline method. Δ indicates the maximum average gap between MINEDRAFT and standard SD or EAGLE. MINEDRAFT consistently outperforms standard SD or EAGLE, reducing latency by up to 20.63% over standard SD and 16.30% over EAGLE.

## C.4. End-to-End Latency Results

**Comparison with baselines.** Fig. 12 shows that MINEDRAFT reduces end-to-end latency by up to 31.13% relative to the best-performing baseline, and the maximum reduction compared to standard SD reaches 38.94%, consistent with the theoretical analysis in Sec. 4 which predicts a minimum improvement of 37%.

**Results with larger draft model.** As shown in Fig. 12, under Setting 4 with Qwen3-8B as the draft model, MINEDRAFT achieves its lowest end-to-end latency when generating 1–2 draft tokens per step. Drafting more than 2 tokens per step results in degraded performance. In this setting, integrating TETRIS does not yield additional reductions in end-to-end latency, as the computational cost of drafting already dominates that of verification. Standard SD fails to execute on this setting due to VRAM constraints. The effect of draft model size on performance is further examined in Sec. C.6.

**Integration with different drafting strategies.** Fig. 13 shows that MINEDRAFT achieves a 15.67% reduction in end-to-end latency relative to the best standard SD result in Setting 5, with a maximum improvement of 20.63%. On the selected EAGLE model settings, MINEDRAFT outperforms the best standalone EAGLE result by up to 6.47% and attains a maximum latency reduction of 16.30%.

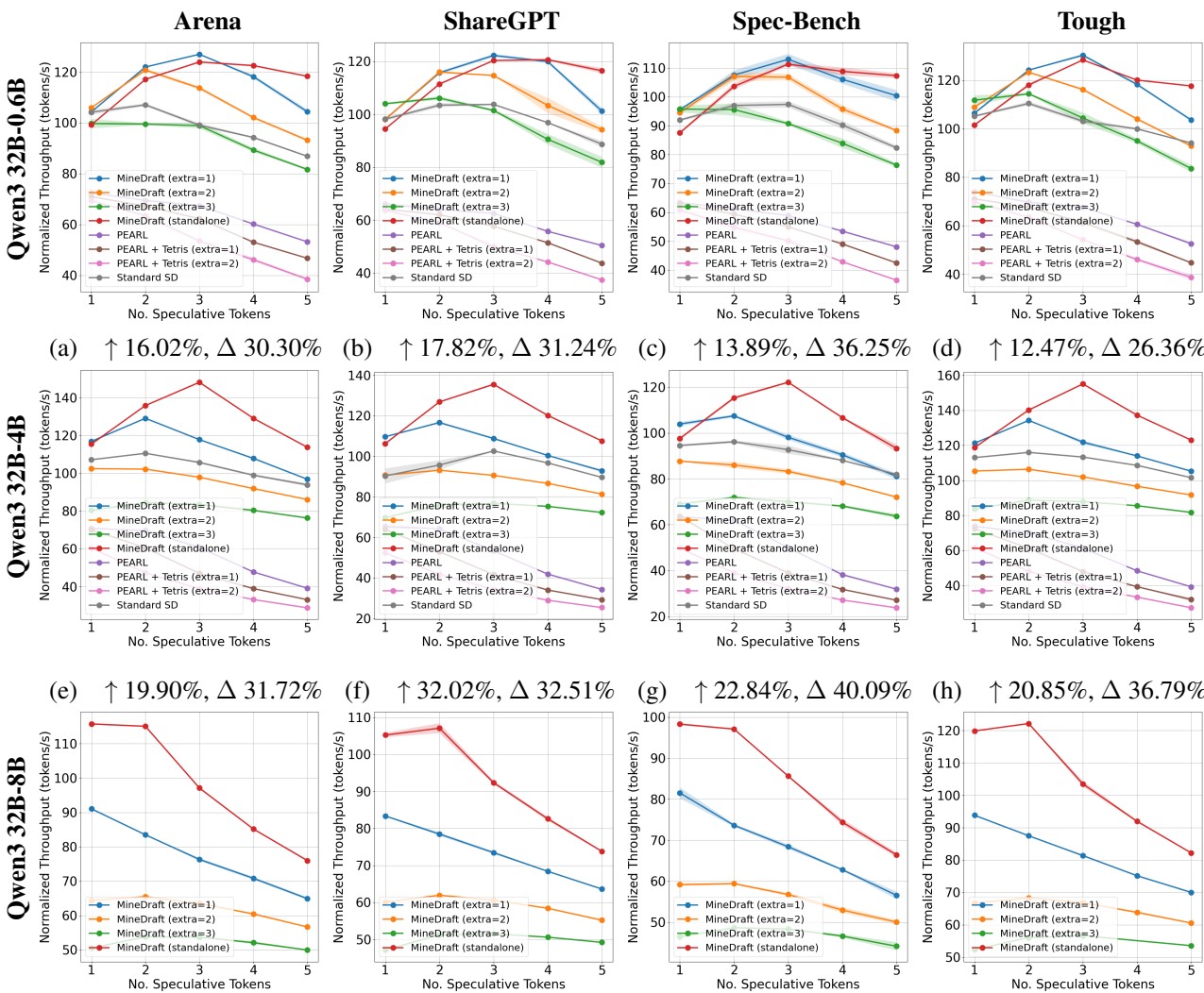

*Figure 14.* Normalized throughput comparison against baseline methods across Settings 1, 3, and 4. ↑ indicates the average improvement over the best baseline method. Δ indicates the maximum average gap between MINEDRAFT and standard SD. Overall, MINEDRAFT improves normalized average throughput by up to 32.02% over the best-performing baseline and by up to 40.09% over standard SD. Standard SD fails in Setting 4 due to OOM.

## C.5. Additional Normalized Results

We present normalized performance comparisons for Settings 1, 3, and 4 on selected datasets. Specifically, we calculate the normalized average throughput by dividing the throughputs of parallel methods by 5, and that of standard SD by 4. End-to-end latency is normalized by multiplying the latencies of parallel methods by 1.25 to account for their proportionally higher compute resource usage relative to standard SD.

Figs. 14 and 15 show that MINEDRAFT outperforms standard SD and PEARL except in regimes where $k$ is small (between 1 and 2), where MINEDRAFT may underperform compared to standard SD. We attribute this degradation in normalized latency to the communication bottleneck inherent in distributed systems (Fernandez et al., 2024), which is well-characterized by the Universal Scalability Law (Gunther, 2008). We empirically validate this hypothesis by profiling experiments across Settings 1–3 on the Spec-Bench dataset using Nsight Systems and computing the ratio of average total communication latency per step to the average end-to-end latency. As reported in Tab. 1, communication overhead dominates the end-to-end latency for small $k$: it accounts for 50.49%–57.93% of the latency when $k = 1$ and 20.96%–34.18% when $k = 2$. For $k > 2$, communication overhead diminishes significantly to between 9.14% and 13.44%. Overall, Fig. 14 demonstrates that MINEDRAFT improves normalized average throughput by up to 32.02% over the best-performing baseline and by up

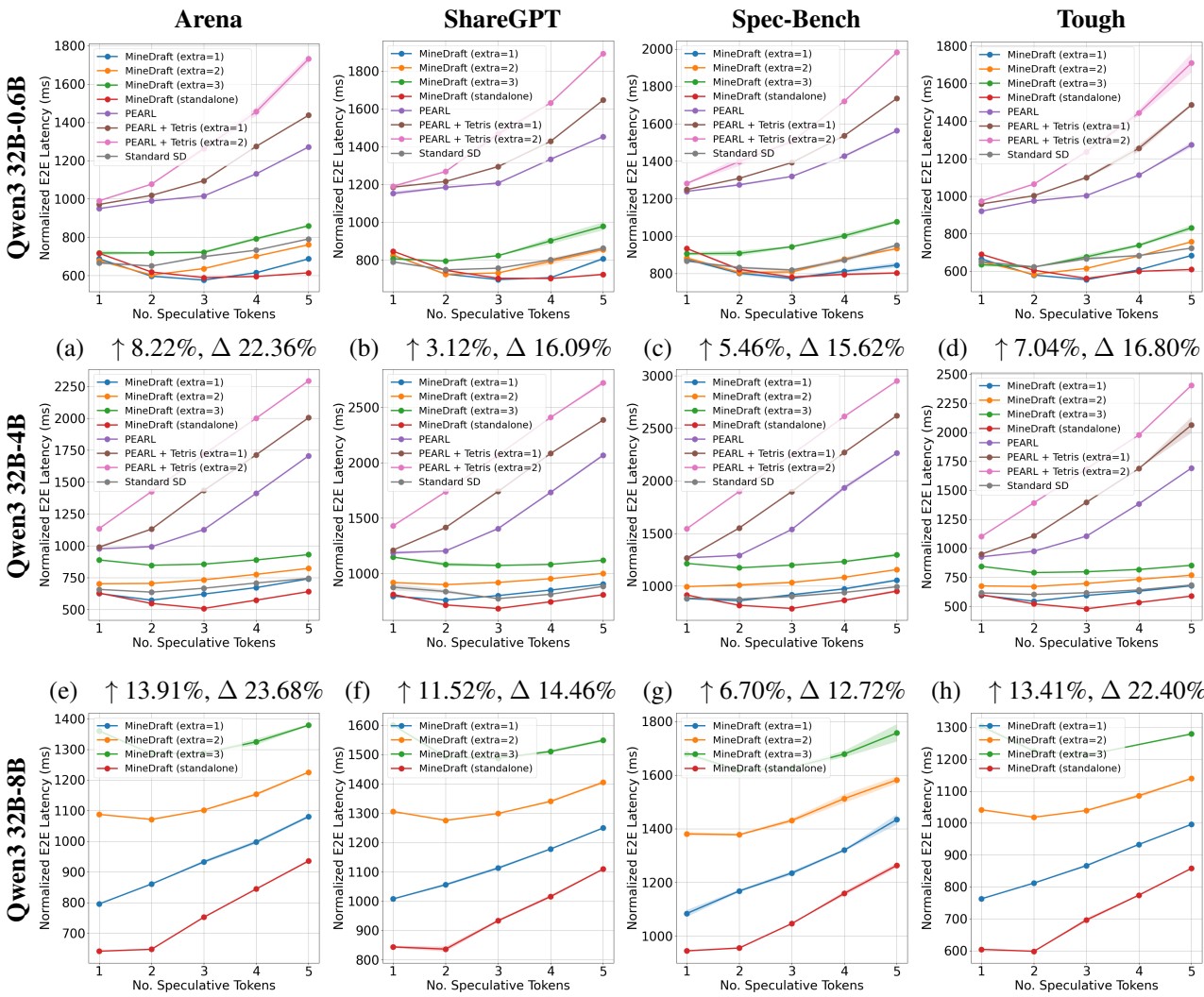

*Figure 15.* Normalized end-to-end latency comparison against baseline methods across Settings 1, 3, and 4. ↑ indicates the average improvement over the best baseline method. Δ indicates the maximum average gap between MineDraft and standard SD. MineDraft outperforms baselines with an exception when $k$ is small ($k = 1$), where MineDraft may be underperformed compared with standard SD. Besides that, MineDraft improves normalized latency by up to 13.91% over the best-performing baseline and by up to 23.68% over standard SD. Standard SD fails in Setting 4 due to OOM.

to 40.55% over standard SD. Similarly, Fig. 15 shows that MineDraft reduces normalized end-to-end latency by up to 13.91% compared to the best baseline results and by up to 24.38% compared to standard SD.

## C.6. Ablation on Draft Model Size

We fix the target model to Qwen3-32B and evaluate MineDraft using three draft models: Qwen3-0.6B, Qwen3-1.7B, and Qwen3-4B. As shown in Fig. 16, performance degrades noticeably when the number of draft tokens per step exceeds 3 for the Qwen3-4B and Qwen3-0.6B draft models, whereas the medium-sized Qwen3-1.7B model is less affected.

These results highlight the need to balance draft quality and speed in MineDraft. Fig. 16 shows that an overly small draft model (e.g., Qwen3-0.6B) may fail to generate sufficiently accurate drafts, leading to a low verification success rate and degraded performance. Conversely, an overly large draft model (e.g., Qwen3-4B), while capable of producing high-quality drafts, may generate them too slowly to benefit from PSD, which requires drafting to be faster than verification. As shown in Fig. 16, Qwen3-1.7B strikes this balance by producing accurate drafts at sufficient speed to exploit parallelized drafting in MineDraft. This observation is supported by Nsight Systems (NVIDIA, 2024) profiling with Spec-Bench in Sec. C.11.

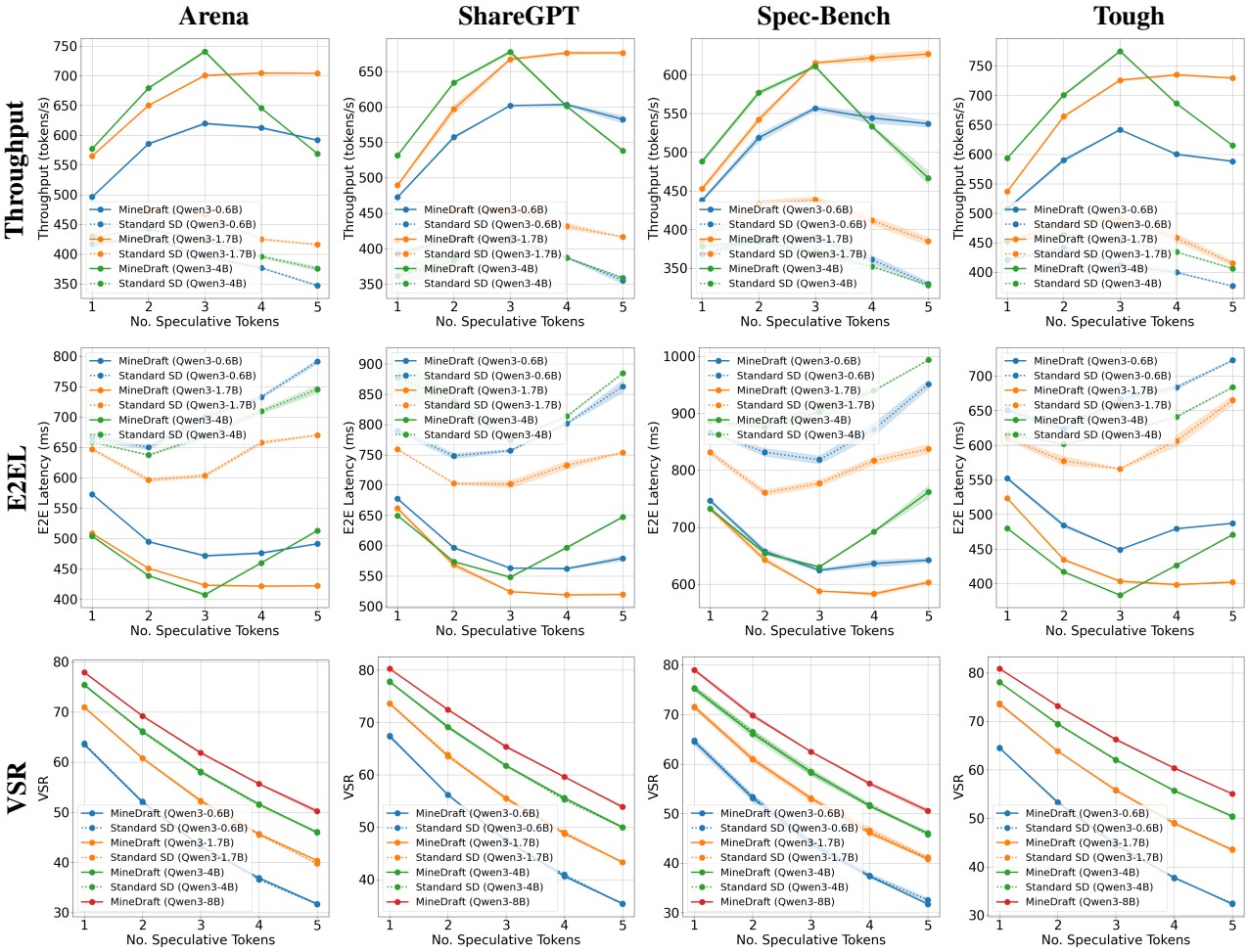

*Figure 16.* Comparisons for various draft models paired with Qwen3-32B using MINEDRAFT. MINEDRAFT always improves average throughput and end-to-end latency compared to standard SD counterparts and matches the VSR of standard SD counterparts across different draft models. More importantly, the performance of Settings 1 (Qwen3-0.6B) and 3 (Qwen3-4B) is degraded when $k > 3$, while that of Setting 2 (Qwen3-1.7B) is not. This phenomenon of degraded performance is discussed in Sec. C.6.

*Table 1.* Percentage of average total communication latency per step over the average end-to-end latency on Spec-Bench across Settings 1–3. Reported numbers are extracted from Nsight Systems profiling results of one run. The percentage of communication latency generally decreases as $k$ increases.

| Setting | $k$ | Percentage of Communication Latency (%) |
|---|---|---|
| Qwen3 32B-0.6B (Setting 1) | 1 | 57.93 |
| | 2 | 34.18 |
| | 3 | 13.44 |
| | 4 | 11.49 |
| Qwen3 32B-1.7B (Setting 2) | 1 | 57.30 |
| | 2 | 26.63 |
| | 3 | 12.74 |
| | 4 | 11.47 |
| Qwen3 32B-4B (Setting 3) | 1 | 50.49 |
| | 2 | 20.96 |
| | 3 | 10.23 |
| | 4 | 9.14 |

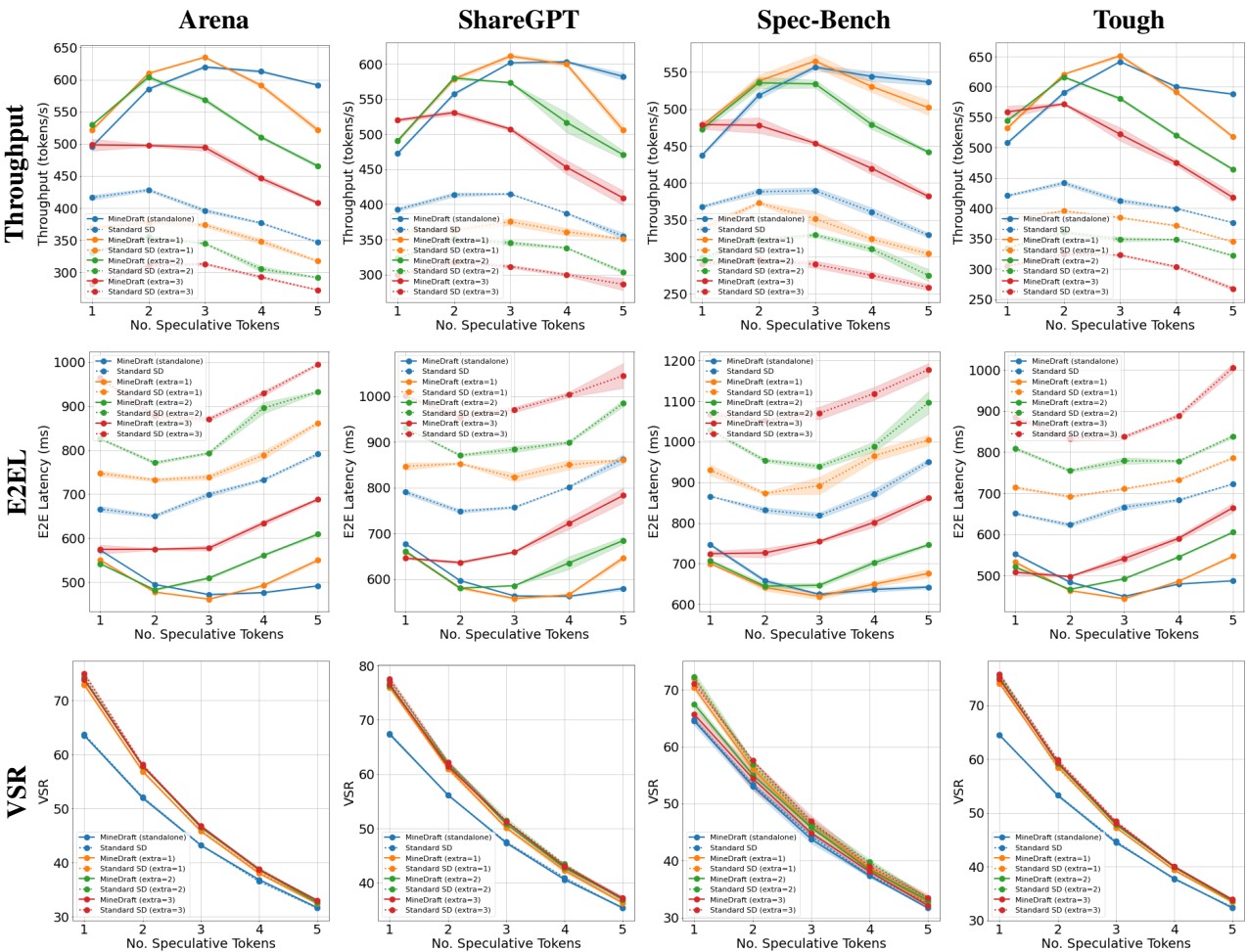

*Figure 17.* Comparisons against standard SD across various $e$ on Setting 1. MINEDRAFT consistently improves average throughput and end-to-end latency over its standard SD counterparts. However, the VSR of MINEDRAFT sometimes degrades relative to standard SD on the Spec-Bench dataset, as analyzed in Sec. C.9.

## C.7. Ablation on Number of Extra Tokens

To evaluate the performance scaling behavior of MINEDRAFT when integrated with TETRIS, we measure the average throughput and end-to-end latency in Setting 1 across varying numbers of extra tokens for TETRIS ($e$). As illustrated in Fig. 17, MINEDRAFT consistently outperforms standard SD across different $e$. However, MINEDRAFT 's integration with TETRIS frequently underperforms relative to standalone MINEDRAFT for $e \geq 3$, as previously noted in Sec. 6.5.

This performance trend mirrors the observations from varying the draft model size in Sec. C.6, where drafting speed and quality both affect overall system performance. Fig. 17 indicates that, while a larger $e$ enables TETRIS to select higher-quality draft tokens for verification and yields a higher verification success rate (VSR), it also prolongs the latency to generate these draft tokens. This trade-off reiterates the need to balance drafting speed and draft quality in MINEDRAFT.

Fig. 17 also reveals that the VSR of MINEDRAFT exhibits little overlap with that of standard SD on the Spec-Bench dataset, whereas on the other datasets, the VSR of MINEDRAFT aligns closely with its standard SD counterparts. This anomaly is analyzed in detail in Sec. C.9.

## C.8. Empirical Results with Varied Sequences per Request and Batch Sizes.

Using Setting 1, we evaluate MINEDRAFT across varying numbers of sequences per request ($n$) and batch sizes ($m$). As shown in Figs. 18 and 19, MINEDRAFT consistently improves both average throughput and end-to-end latency over standard

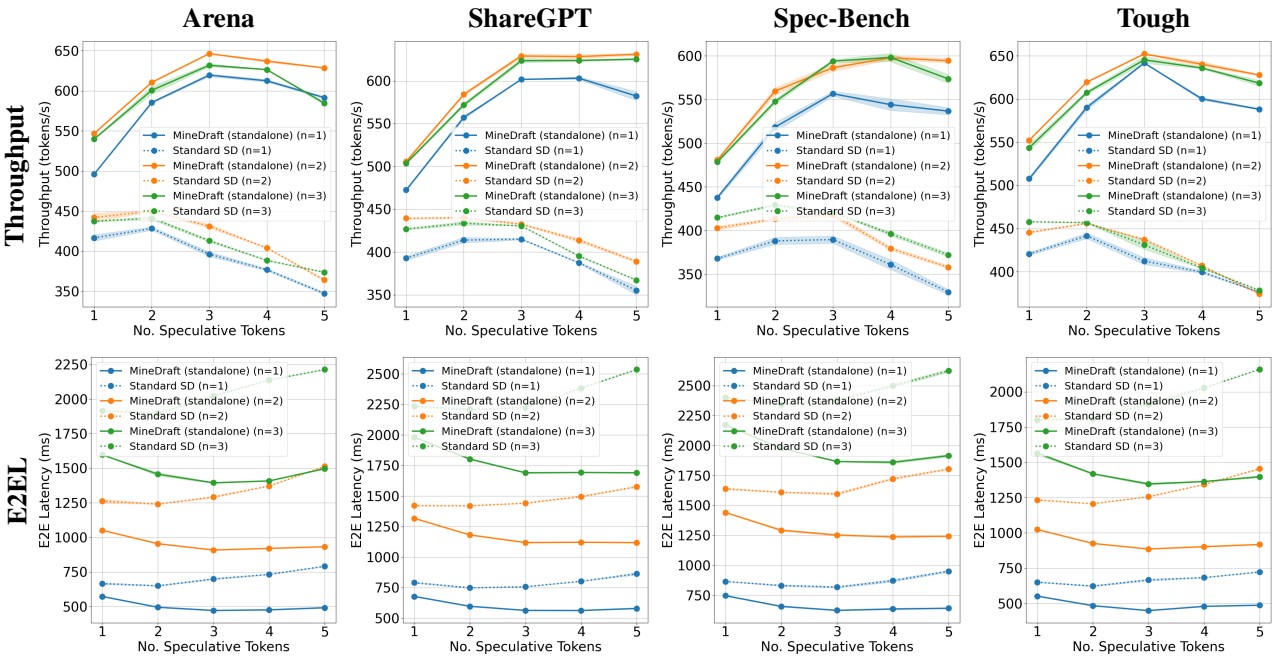

*Figure 18.* Throughput and end-to-end latency comparisons against standard SD across various $n$ on Setting 1. MINEDRAFT consistently outperforms standard SD across all selected $n$ values. Throughput of MINEDRAFT reaches the maximum when $n = 2$ except in the Spec-Bench dataset, where $n = 3$ may also take the maximum throughput at specific $k$ values (3 and 4).

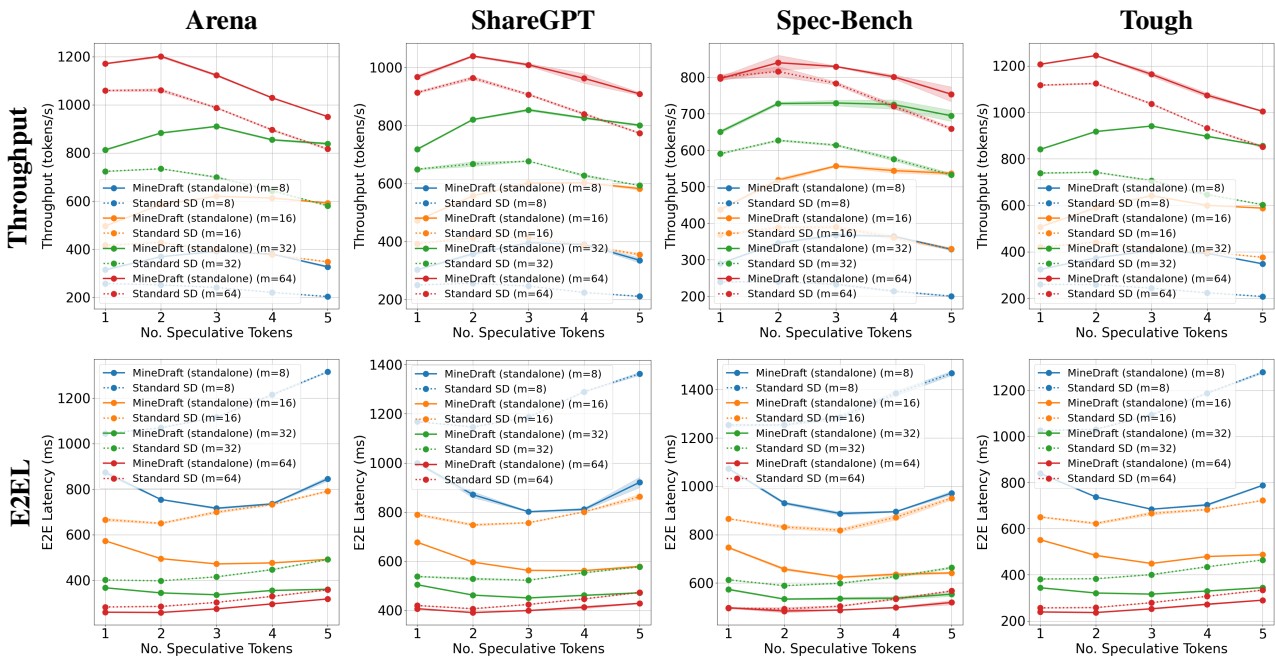

*Figure 19.* Throughput and end-to-end latency comparisons against standard SD across various $m$ (batch size) on Setting 1. The plots show that the performance of MINEDRAFT is sometimes comparable to that of standard SD with a doubled batch size ($2m$) at a large $k$ value (4 to 5).

SD across all tested configurations of $n$ and $m$. Specifically, varying $n$, MINEDRAFT achieves up to 72.64% improvement in throughput and 38.41% reduction in latency; varying $m$, the corresponding improvements reach 75.81% and 40.79%.

As shown in Fig. 19, when the number of draft tokens per step reaches five, the performance of MINEDRAFT becomes

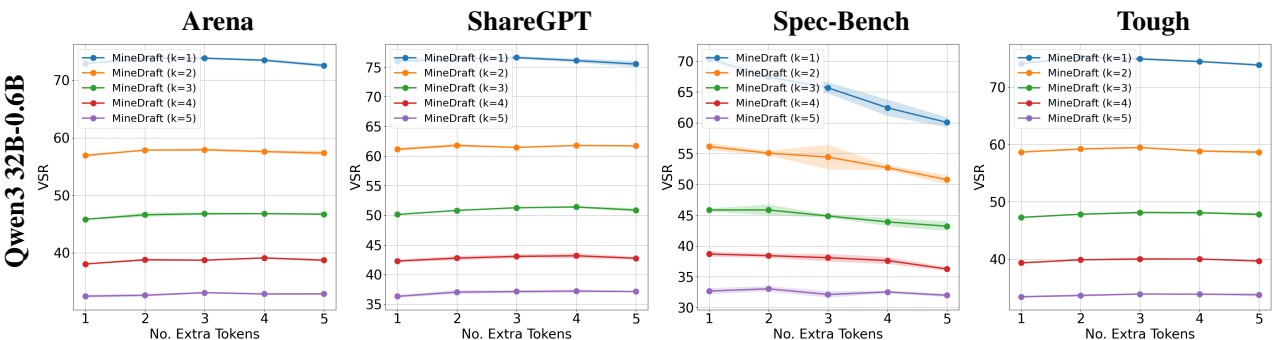

*Figure 20.* VSR comparison for MINEDRAFT with TETRIS across extra tokens on Setting 1. VSR of MINEDRAFT 's integration with TETRIS generally increases as the number of extra tokens ($e$) increases except on Spec-Bench.

comparable to that of standard SD with a doubled batch size ($2m$). This result highlights the effectiveness of MINEDRAFT as a batch PSD framework: with appropriate adaptive drafting strategies and model configurations, MINEDRAFT can further optimize performance even when a large number of draft tokens are generated per step.

### C.9. Analysis of Verification Success Rate

The verification success rate (VSR) is defined as follows:

$$VSR = \frac{\textit{Accepted tokens}}{\textit{Tokens sent for verification}},$$

which is the proportion of tokens successfully verified out of the total number of tokens submitted for verification. VSR measures the quality of the selected draft tokens. We analyze the VSR of MINEDRAFT integrated with TETRIS under Settings 1 and 5. As shown in Figs. 20 and 22, VSR generally increases with the number of extra tokens, except on the Spec-Bench dataset, where VSR decreases more frequently than on the other datasets.

We attribute this anomaly to the combined effects of Spec-Bench's smaller sample size and TETRIS's behavior on non-full batches. Because Spec-Bench contains fewer conversations than the other datasets, its requests are exhausted more quickly, causing both batches to fall below their nominal capacity $m$ earlier in the lifespan of MINEDRAFT. When TETRIS operates on non-full batches, the reduced pool of candidate draft tokens increases the likelihood of selecting lower-quality drafts, thereby raising the target model's rejection rate. As a result, VSR decreases during non-full-batch operation; since a larger fraction of SD steps occur under this regime, the aggregate VSR is negatively affected, effectively diluting the high VSR observed during the initial, fully saturated steps.

To validate this explanation for the VSR degradation observed on Spec-Bench, we compute VSR over truncated portions of the execution, considering only the initial 70%–100% or 80%–100% of SD steps. As shown in Figs. 21 and 23, on Spec-Bench, the VSR of MINEDRAFT integrated with TETRIS remains comparable to or higher than that of standalone TETRIS until approximately 82–86% of SD steps for $n = 1$, and until approximately 92% for $n = 2$. In contrast, on the Arena, ShareGPT, and Tough datasets, the VSR of MINEDRAFT with TETRIS matches that of standalone TETRIS over the first 98% of SD steps for $n = 1$, and throughout the entire evaluation for $n = 2$.

### C.10. Additional Experiment Results on Larger Models

We conduct additional experiments on a large Mixture-of-Experts (MoE) target model (Shazeer et al., 2017; Dai et al., 2024; Yang et al., 2025), Qwen3-235B-A22B-Instruct-2507-FP8 (Yang et al., 2025), to evaluate the compatibility of MINEDRAFT with modern MoE architectures. We use Qwen3-14B (Yang et al., 2025) as the draft model with $m = 64$, $n = 2$, a temperature of $0.8$, and served on five H100 GPUs. Under this configuration, standard SD is infeasible due to VRAM constraints. As shown in Fig. 24, MINEDRAFT achieves its best performance when four draft tokens are generated per step, reaching an average throughput of approximately 1200–1600 tokens per second.

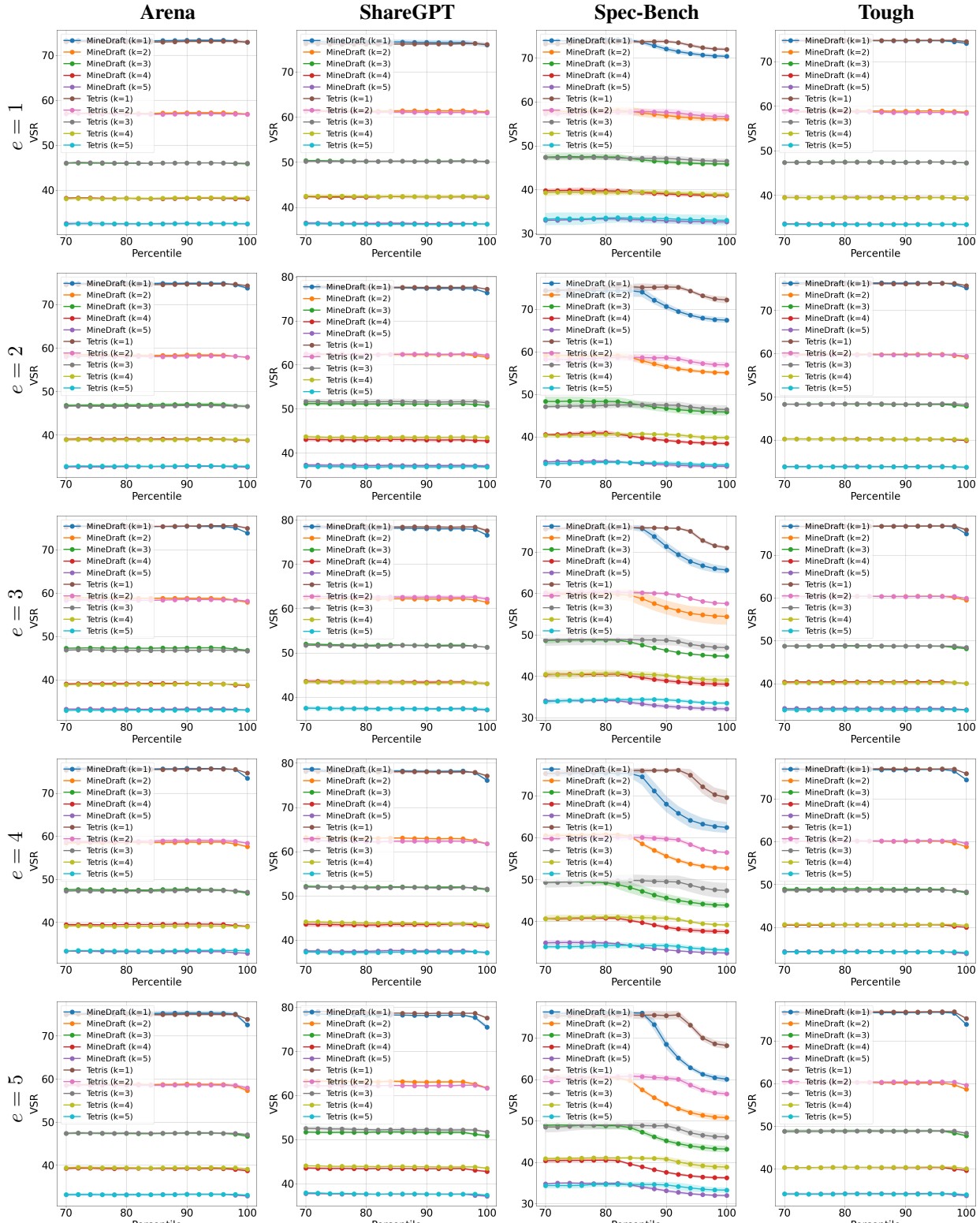

*Figure 21.* VSR comparison for MINEDRAFT with TETRIS and standalone TETRIS across trimming percentiles on Setting 1. VSR of MINEDRAFT 's integration with TETRIS becomes worse than that of standalone TETRIS from approximately 82–86% of SD steps on Spec-Bench. On the remaining datasets, VSR of MINEDRAFT integrated with TETRIS is consistent with that of standalone TETRIS over the first 98% of SD steps.

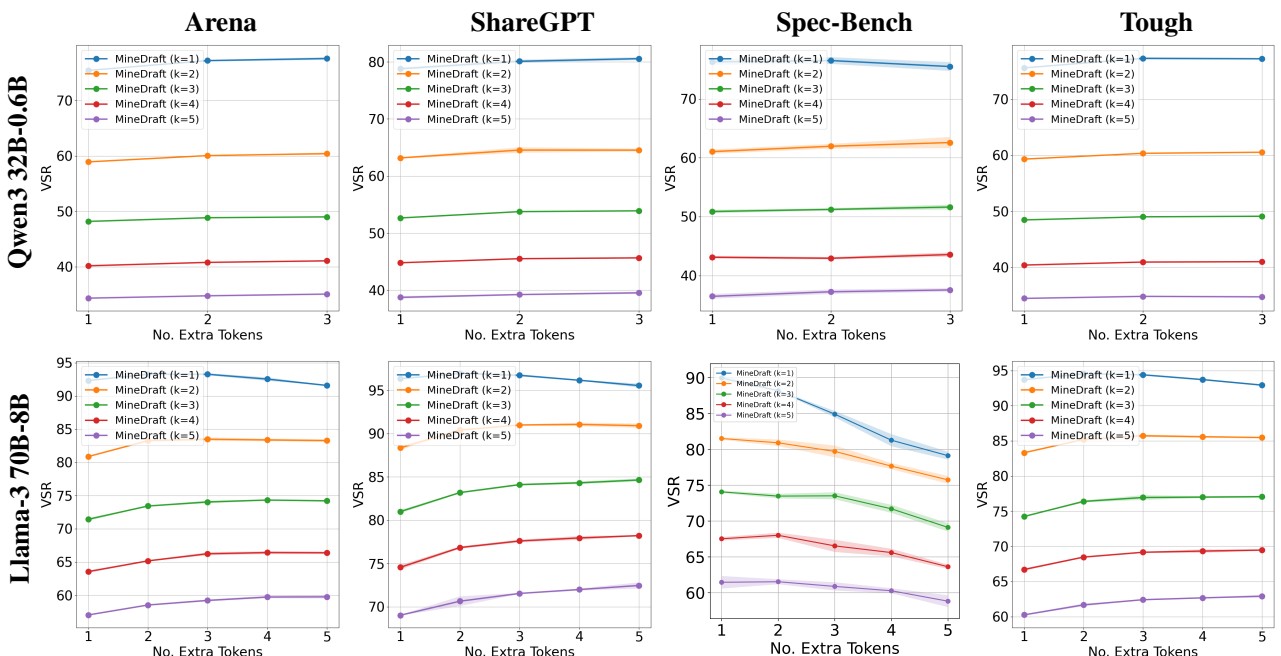

*Figure 22.* VSR comparison for MINEDRAFT with TETRIS across Settings 1 and 5 on $n = 2$ and temperature $= 0.8$. VSR of MINEDRAFT 's integration with TETRIS generally increases as the number of extra tokens ($e$) increases except on Spec-Bench.

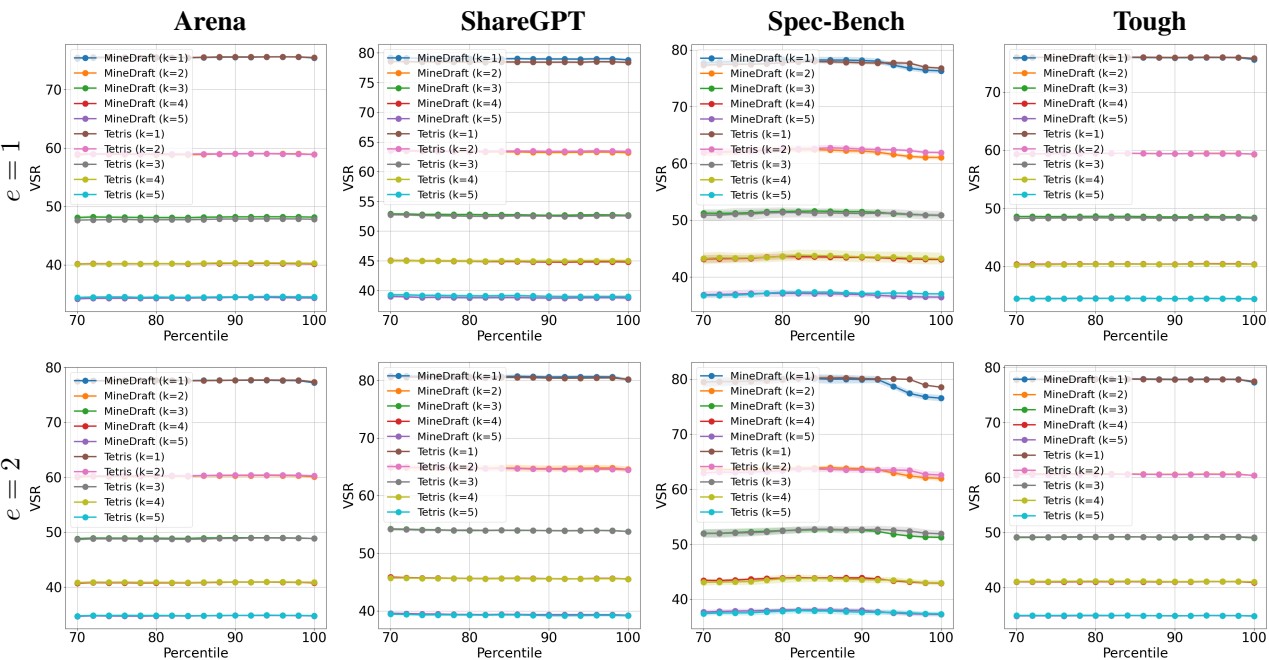

*Figure 23.* VSR comparison for MINEDRAFT with TETRIS and standalone TETRIS across trimming percentiles on Setting 1, $n = 2$ and temperature $= 0.8$. VSR of MINEDRAFT 's integration with TETRIS becomes worse than that of standalone TETRIS from approximately 92% of SD steps on Spec-Bench. On the remaining datasets, VSR of MINEDRAFT integrated with TETRIS is consistent with that of standalone TETRIS throughout the experiments.

## C.11. Nsight Systems Profiling Results

Nsight Systems profiling in Settings 1–3 using the Spec-Bench dataset shows that Qwen3-1.7B and Qwen3-0.6B exhibit similar drafting times, whereas Qwen3-4B has the longest drafting time among the three draft models. As illustrated in

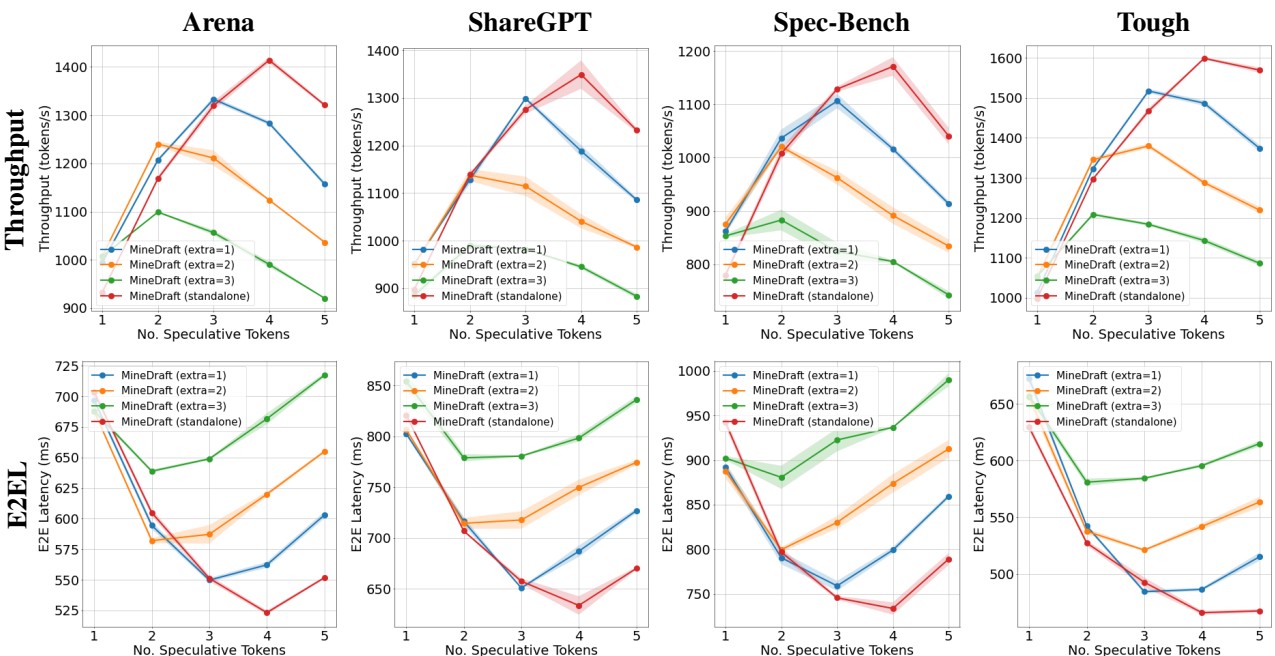

*Figure 24.* MINEDRAFT throughput and end-to-end latency results on Qwen3-235B-A22B-Instruct-2507-FP8 paired with Qwen3-14B, $n = 2$ and temperature $= 0.8$. The reported numbers reflect the mean and standard deviation over 2 independent trials. MINEDRAFT reaches the optimal performance when $k = 4$, producing an average throughput of 1200-1600 tokens/s. Standard SD fails in this setting due to OOM.

Fig. 25, Qwen3-4B spends significantly more time on drafting than on verification, limiting its ability to benefit from PSD. In contrast, Qwen3-1.7B, while slightly slower than Qwen3-0.6B, has greater model capacity and is thus more effective at generating draft tokens that pass verification. This observation supports the conclusion in Sec. C.6 that Qwen3-1.7B achieves the optimal balance between drafting speed and draft quality.

### C.12. Parallelization Coverage

We assess the impact of the tail effect discussed in Sec. A by quantifying the proportion of decoding steps executed in parallel mode. Tab. 2 reports these rates across various model settings and datasets. The results show that MINEDRAFT achieves an overall average parallelization rate of 99.19%, indicating that the limitation imposed by the tail effect is negligible.

## D. Frequently Asked Questions

**Question 1.** Why does MINEDRAFT not attain the maximum theoretical latency improvement of 50% in the experiments?

**Answer.** The theoretical upper bound of a 50% latency reduction assumes an idealized setting in which drafting and verification times are perfectly matched throughout the execution of MINEDRAFT. In practice, both stages exhibit stochastic variation due to system overheads, resource contention, and time multiplexing, making the precise temporal alignment required to attain this bound infeasible in real-world deployments.

**Question 2.** Why does MINEDRAFT maintain two batches instead of more? Why is the "Batch Parallelism size" fixed to 2?

**Answer.** The design of MINEDRAFT is intrinsically tied to the two-stage structure of speculative decoding (SD): drafting and verification. Compared to autoregressive decoding, the additional drafting stage introduces extra latency, which can become prohibitive when either the number of draft tokens per step is large or the draft model is computationally expensive. MINEDRAFT mitigates this overhead by overlapping the drafting of one set of requests with the verification of another. This design naturally requires exactly two batches: one for drafting and one for verification. Introducing additional batches would not produce further parallelism benefits, as there are no additional stages to overlap, and would instead incur unnecessary scheduling and management overhead.

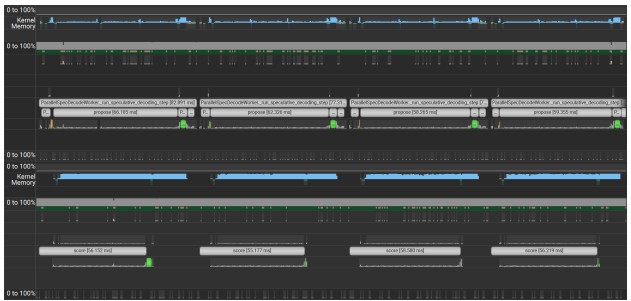

**(a)** Sample segment of Nsight Systems profiling results with Setting 1, Spec-Bench dataset, and $k = 3$. Median drafting and verification times are 49.60 ms and 48.62 ms, respectively, suggesting MINEDRAFT almost perfectly overlaps drafting with verification under this configuration.

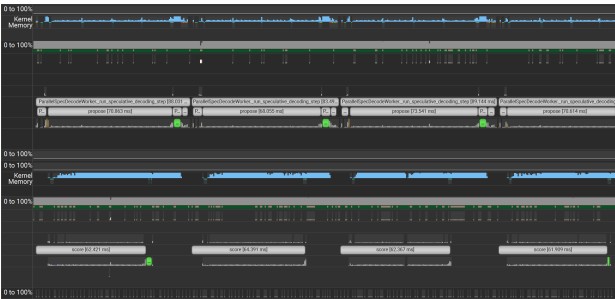

**(b)** Sample segment of Nsight Systems profiling results with Setting 1, Spec-Bench dataset, and $k = 4$. Median drafting and verification times are 64.06 ms and 51.06 ms, respectively, suggesting drafting time exceeds verification time by 25.48%, and the performance of MINEDRAFT may degrade under this configuration.

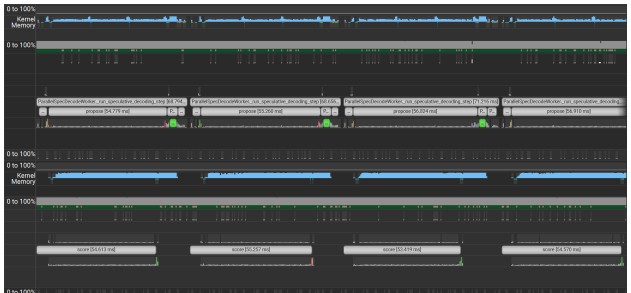

**(c)** Sample segment of Nsight Systems profiling results with Setting 2, Spec-Bench dataset, and $k = 3$. Median drafting and verification times are 49.53 ms and 49.30 ms, respectively, suggesting MINEDRAFT almost perfectly overlaps drafting with verification under this configuration.

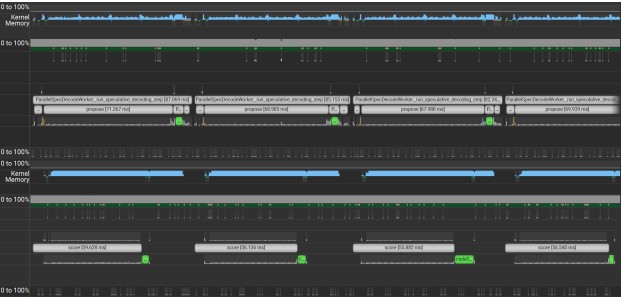

**(d)** Sample segment of Nsight Systems profiling results with Setting 2, Spec-Bench dataset, and $k = 4$. Median drafting and verification times are 65.75 ms and 50.70 ms, respectively, suggesting drafting time exceeds verification time by 29.68%, and the performance of MINEDRAFT may degrade under this configuration.

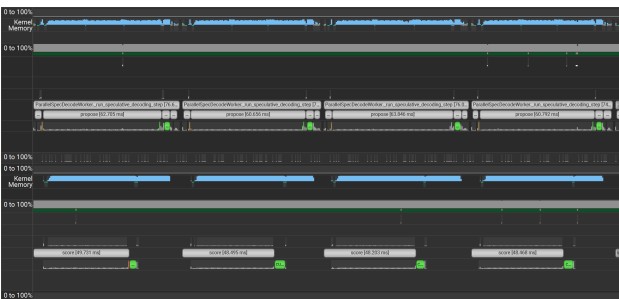

**(e)** Sample segment of Nsight Systems profiling results with Setting 3, Spec-Bench dataset, and $k = 3$. Median drafting and verification times are 61.83 ms and 47.62 ms, respectively, suggesting drafting time exceeds verification time by 29.86%, and the performance of MINEDRAFT may degrade under this configuration.

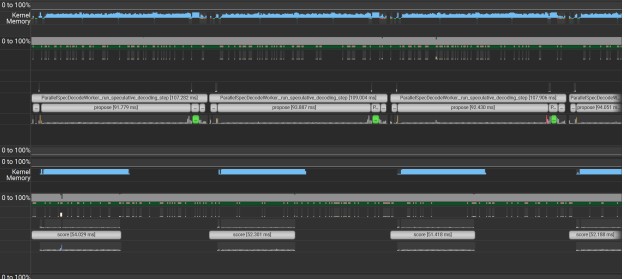

**(f)** Sample segment of Nsight Systems profiling results with Setting 3, Spec-Bench dataset, and $k = 4$. Median drafting and verification times are 81.00 ms and 48.93 ms, respectively, suggesting drafting time exceeds verification time by 65.52%, and the performance of MINEDRAFT is highly likely to degrade under this configuration.

*Figure 25.* Comparison of Nsight Systems profiling across 6 different configurations.

**Question 3.** Why does MINEDRAFT not incorporate prefill requests in the parallelism?

*Table 2.* Percentage of total steps when standalone MineDraft operates in parallel vs. sequential mode across different settings and datasets. Reported numbers are averaged from three independent runs. MineDraft achieves an average parallelization percentage of 99.19%.

| Setting | $k$ | Arena | ShareGPT | Spec-Bench | Tough |
|---|---|---|---|---|---|
| Qwen3 32B-0.6B (Setting 1) | 1 | 99.84 | 99.96 | 94.32 | 99.88 |
| | 2 | 99.78 | 99.83 | 98.10 | 99.85 |
| | 3 | 99.86 | 99.84 | 98.61 | 99.76 |
| | 4 | 99.80 | 99.81 | 98.85 | 99.93 |
| | 5 | 99.86 | 99.61 | 98.79 | 99.87 |
| Qwen3 32B-1.7B (Setting 2) | 1 | 99.90 | 99.75 | 93.61 | 99.94 |
| | 2 | 99.94 | 99.77 | 97.16 | 99.92 |
| | 3 | 99.72 | 99.79 | 98.13 | 99.81 |
| | 4 | 99.60 | 99.75 | 98.82 | 99.86 |
| | 5 | 99.78 | 99.66 | 98.02 | 99.86 |
| Qwen3 32B-4B (Setting 3) | 1 | 99.96 | 99.86 | 93.00 | 99.90 |
| | 2 | 99.90 | 99.83 | 95.66 | 99.96 |
| | 3 | 99.89 | 99.81 | 97.52 | 99.88 |
| | 4 | 99.83 | 99.65 | 98.51 | 99.84 |
| | 5 | 99.83 | 99.71 | 99.31 | 99.73 |
| Qwen3 32B-8B (Setting 4) | 1 | 99.93 | 99.95 | 93.48 | 99.94 |
| | 2 | 99.91 | 99.88 | 95.87 | 99.94 |
| | 3 | 99.95 | 99.70 | 98.91 | 99.81 |
| | 4 | 99.83 | 99.88 | 97.12 | 99.85 |
| | 5 | 99.65 | 99.83 | 98.16 | 99.74 |
| Llama-3 70B-8B (Setting 5) | 1 | 99.77 | 99.88 | 99.59 | 99.81 |
| | 2 | 99.68 | 99.71 | 98.01 | 99.80 |
| | 3 | 99.64 | 99.60 | 97.02 | 99.81 |
| | 4 | 99.50 | 99.19 | 98.05 | 99.73 |
| | 5 | 98.86 | 99.53 | 95.62 | 99.63 |
| Vicuna 33B-EAGLE (Setting 6) | 1 | 99.89 | 99.81 | 98.59 | 99.76 |
| | 2 | 99.78 | 99.69 | 97.63 | 99.91 |
| | 3 | 99.89 | 99.63 | 96.36 | 99.82 |
| | 4 | 99.86 | 99.72 | 98.58 | 99.71 |
| | 5 | 99.88 | 99.76 | 99.56 | 99.82 |
| Vicuna 13B-EAGLE (Setting 7) | 1 | 99.89 | 99.91 | 96.01 | 99.92 |
| | 2 | 99.88 | 99.74 | 97.31 | 99.87 |
| | 3 | 99.75 | 99.87 | 99.57 | 99.83 |
| | 4 | 99.73 | 99.88 | 97.52 | 99.83 |
| | 5 | 99.76 | 99.68 | 98.18 | 99.75 |

**Answer.** The current design of MINEDRAFT assumes that chunked prefill is disabled. Under this configuration, vLLM completes the prefill stage for new requests in a dedicated step before transitioning them to decoding in subsequent steps, isolating prefill requests from the speculative decoding steps in which MINEDRAFT operates. As support for chunked prefill is a planned extension, future versions of MINEDRAFT will integrate prefill requests into its parallel execution workflow.

