# OpenReview forum: "MineDraft: A Framework for Batch Parallel Speculative Decoding"
_ICML.cc/2026/Conference — ICML 2026 regular_

### Official Review · Reviewer_sqab · 2026-03-05

**Soundness:** 3
**Presentation:** 3
**Significance:** 3
**Originality:** 3
**Overall Recommendation:** 4
**Confidence:** 4

**Summary:**

This paper proposes MINEDRAFT, a framework for Batch Parallel Speculative Decoding (PSD) that overlaps the drafting and verification stages of speculative decoding at the batch level. The key idea is to maintain two alternating batches so that the draft model generates tokens for one batch while the target model verifies tokens for the other batch, thereby hiding drafting latency. The framework includes a batch manager and scheduler that integrate with vLLM to support continuous batching and efficient KV cache allocation. The paper also presents a simple latency model comparing sequential speculative decoding and the proposed parallel approach. Experiments on several LLMs demonstrate improvements in both latency and throughput.

**Compliance With Llm Reviewing Policy:**

Affirmed.

**Final Justification:**

The authors have adequately addressed the concerns raised in my initial review during the rebuttal phase. I therefore maintain my positive score.

**Key Questions For Authors:**

1. How sensitive is the performance to the drafting model size or acceptance rate?
2. Can the approach generalize to multi-node inference settings where communication overhead becomes significant?

**Limitations:**

Yes.

**Strengths And Weaknesses:**

**Strengths**

- The paper identifies an important inefficiency in speculative decoding systems: the sequential execution of drafting and verification. The proposed batch-level pipelining is simple yet effective for hiding drafting latency.
- The work highlights that speculative decoding performance is often limited by runtime scheduling rather than model accuracy, which is an important perspective for the community.
- The framework is orthogonal to many speculative decoding variants (e.g., EAGLE, TETRIS) and can be combined with them, increasing its practical value.
- The integration with vLLM and support for continuous batching and PagedAttention make the approach realistic for deployment.
- Experiments show meaningful gains in throughput and latency across several models and configurations.

**Weaknesses**

- The core idea is essentially a pipeline parallelization strategy between drafting and verification stages. While effective, the conceptual novelty from an ML perspective is somewhat limited.
- The framework uses a two-batch pipeline, but the paper does not explore whether more flexible or adaptive scheduling strategies could further improve utilization.

---

> ### Author Rebuttal · Authors · 2026-03-31
>
> Thank you for your positive review. We appreciate your recognition that our Batch Parallelism **"is simple yet effective for hiding drafting latency"** and that our experiments **demonstrate meaningful throughput and latency gains** across models and settings. We also value your endorsement of MineDraft's **orthogonality to existing drafting methods**, its **practical integration with vLLM**, and its **importance as a perspective for improving speculative decoding performance**. For clarity, we group related weaknesses and questions and address them as follows:
>
> > **Limited conceptual novelty**
>
> Although the idea of parallel speculative decoding is intuitively effective, it is inherent with the following core challenges that MineDraft resolves:
>
> 1. **Breaking the dependency of verification on drafting with reasonable additional GPU resources.** Existing methods like Minions (Wang et al., 2024) and DSI (Timor et al., 2025) may require substantial additional GPU resources to operate in maximum efficiency (Lines 34 - 38, right column). **MineDraft requires only 1 additional GPU to achieve parallel SD, balancing GPU resources and performance improvements.** The comparisons of resource requirement between MineDraft and these methods are provided in Table 1 at this anonymized link: https://anonymous.4open.science/r/anon-F221/tbl1.jpg.
> 2. **Wide practical applicability to existing drafting methods and models.** Prior work like ParallelSpec (Xiao et al., 2024) requires a dedicated training stage for the draft model (Lines 39 - 40, right column), limiting its practical applicability. **MineDraft is agnostic to draft generation and selection strategies and orthogonal to most LLM inference optimization techniques** (Lines 77 - 83, left column), achieving a significant practical impact.
> 3. **Scalability to batched request settings.** While some methods like PEARL (Liu et al., 2025) are effective in single request settings, their performance may degrade when scaled to batched request settings where multiple requests are simultaneously served (Lines 44 - 46, right column). **MineDraft is designed to handle batched request settings and operate with an efficient, continuous batching strategy**, providing hassle-free scalability for batched request serving.
>
> > **The framework uses a two-batch pipeline, but the paper does not explore whether more flexible or adaptive scheduling strategies could further improve utilization.**
>
> We assume the reviewer means *adaptive drafting methods* that dynamically adjust the number of draft tokens, like BanditSpec (Hou et al., 2025), TETRIS (Wu et al., 2025), and SpecServe (Huang et al., 2025), using "flexible or adaptive scheduling strategies". These methods would indeed **sometimes further improve throughput and utilization, and we have provided benchmark results for TETRIS integration with MineDraft in the paper**. Analyzing performance differences between standalone MineDraft and integrations with such drafting methods is an interesting future direction (Lines 415 - 417, left column).
>
> > **Performance sensitivity to the drafting model size or acceptance rate?**
>
> Thank the reviewer for questioning MineDraft's performance sensitivity to draft model size and acceptance rate. **The performance sensitivity to draft model size has been shown jointly in Sections C.5 (Fig. 13) and C.9 (Fig. 25).** Specifically, the empirical results reveal that standalone MineDraft prefers a draft model that generates accurate drafts at a sufficient speed, aligning with verification time, thereby achieving a balance between acceptance rate and the draft-verify overlapping effect. Regarding sensitivity to the acceptance rate (Verification Success Rate, or VSR), we would like to underscore that **VSR is correlated with draft model size; a larger draft model implies a higher VSR**. We provided VSR comparisons among different draft models paired with Qwen3-32B in Figure 28 at this anonymized link: https://anonymous.4open.science/r/anon-F221/fig28.jpg. However, **larger draft models also imply longer drafting times, sometimes undermining the parallelization effect and performance gains** (shown in Fig. 25).
>
>
> > **Can the approach generalize to multi-node inference settings?**
>
> We appreciate the reviewer for raising the question on MineDraft's scalability to multi-node deployments. **Since MineDraft supports model parallelism methods for target models like tensor parallelism, it can be generalized to multi-node inference settings by configuring vLLM.** The evaluation of multi-node deployments is left for future work due to resource constraints.
>
> *We sincerely hope our response has resolved your concerns. We are also grateful for any additional comments you may have; if you have any further concerns we missed in our response, please let us know. If our response helped clarify our paper and resolve your questions, we would be deeply grateful for your openness in adjusting your scores accordingly.*

---

> > ### Author Rebuttal · Reviewer_sqab · 2026-04-01
> >
> > The authors have adequately addressed most of my concerns in the rebuttal. I will maintain my positive score.

---

> > > ### Author Response · Authors · 2026-04-08
> > >
> > > Thank you for the careful review of our paper and constructive feedback. We are glad that our responses have addressed your concerns. Your feedback will help us further improve the positioning of our paper.
> > >
> > > Given that your concerns have now been addressed, we would greatly appreciate your consideration in revisiting your *Overall Recommendation* and continuing to support our paper in the subsequent discussion.

---

### Official Review · Reviewer_nCd7 · 2026-03-12

**Soundness:** 3
**Presentation:** 3
**Significance:** 3
**Originality:** 3
**Overall Recommendation:** 5
**Confidence:** 3

**Summary:**

MineDraft proposes batch parallel speculative decoding (PSD): overlap drafting for one batch of requests with verification for another by maintaining two alternating batches and placing the draft model on a separate GPU. Theoretical analysis shows PSD achieves ≥37% latency reduction over standard SD under mild assumptions. Implemented as a vLLM plugin w/ PagedAttention compatibility. Eval'd across 7 model configs (Qwen3, Llama-3.3-70B, Vicuna w/ EAGLE). Up to 75% throughput gain and 39% E2EL reduction over standard SD at cost of 1 extra GPU.

**Compliance With Llm Reviewing Policy:**

Affirmed.

**Key Questions For Authors:**

1. What's the throughput-per-GPU comparison? MineDraft uses 5 GPUs vs 4 for standard SD. Does the 75% throughput gain survive normalization to equal hardware?
2. How often does the tail effect / fallback to standard SD occur in practice? What fraction of total steps run in parallel mode vs fallback?
3. How does MineDraft handle variable output lengths? Fixed 256 tokens is unrealistic. When requests in one batch finish much earlier, does batch imbalance become severe?
4. How sensitive is Thm 1 to the parametric form f(t)=1-e^{-αt}? Does the 37% bound hold for other realistic Pareto frontiers?

**Limitations:**

Discussed in Sec A (batch imbalance, tail effect, chunked prefill). Honest but could quantify the tail effect's impact. Extra GPU cost not discussed as a limitation per se.

**Strengths And Weaknesses:**

Strengths:

- simple, clean idea. overlapping draft/verify via two alternating batches is intuitive and well-executed
- theoretical result (Thm 1) is non-trivial and informative. 37% minimum improvement under mild assumptions on Pareto frontier gives useful guidance
- very strong engineering contribution. vLLM plugin w/ PagedAttention, continuous batching, KV block management patches. production-ready
- comprehensive eval: 7 model settings, 4 datasets, integration w/ EAGLE and TETRIS, ablations on draft model size / batch size / n. Nsight profiling is a nice touch
- orthogonal to existing SD methods. can compose w/ EAGLE, TETRIS, etc. and shows additive gains
- practical benefit: also alleviates VRAM contention by placing draft model on separate GPU (Fig 5, Setting 4 where standard SD OOMs)

Weaknesses:

- requires 1 extra GPU. for the Qwen3-32B settings that's 5 GPUs vs 4 (25% more hardware). throughput-per-GPU comparison would be more informative than raw throughput
- the two-batch design introduces a "tail effect" when one batch empties and system falls back to standard SD. paper acknowledges this but doesn't quantify how often it occurs or its impact on realistic continuous-serving workloads
- no comparison w/ DSI (Timor et al., 2025) or Minions (Wang et al., 2024), which also parallelize SD. paper cites them but only as "requiring more resources." a direct comparison would clarify the tradeoff
- chunked prefill not supported. acknowledged as future work, but this is standard in production vLLM. limits practical applicability
- theory assumes f(t) = 1-e^{-αt}, a specific parametric form. how sensitive is the 37% bound to this assumption? no analysis of robustness to different Pareto frontier shapes
- batch imbalance issues (Sec A) seem fragile. preemption, abort, variable request lengths can all disrupt balance. the mitigation ("assign to draft batch") has its own overhead
- EAGLE performance degradation noted as a vLLM implementation issue (Sec 6.2). makes EAGLE integration results hard to interpret
fixed output length (256 tokens) in all experiments. production workloads have highly variable output lengths. how does MineDraft handle heterogeneous completion times across the two batches?

---

> ### Author Rebuttal · Authors · 2026-03-31
>
> Thank you for recognizing that our Batch Parallelism is a **"simple, clean, intuitive, and well-executed"** idea, and that our Theorem 1 is **"non-trivial and informative"**. We also appreciate your acknowledgement of our **"strong, production-ready engineering contribution"** and **"comprehensive evaluation"** demonstrating **"additive gains"** with existing SD methods. We further value your comments that the **"Nsight profiling"** is insightful, Theorem 1 provides **"useful guidance"**, and MineDraft **"alleviates VRAM contention"** via a separate GPU for drafting. For clarity, we group related weaknesses and questions and address them as follows:
>
> > **Comparison with DSI and Minions**
>
> Please kindly refer to the comparisons of resource requirements with Minions (Wang et al., 2024) and DSI (Timor et al., 2025) in Table 1 at this anonymized link: https://anonymous.4open.science/r/anon-F221/tbl1.jpg.
>
> > **Chunked prefill**
>
> Yes, chunked prefill is not supported in the current version of MineDraft; it is an interesting future direction (Lines 433 - 435, right column).
>
> > **Batch imbalance issues**
>
> We would like to clarify that "assign to draft batch" is not our proposed mitigation to resolve the batch imbalance issue described in Sec. A. The true mitigation is to apply the `assign` operation, as specified in Sec. 5.1 (Lines 194 - 203, right column), whenever assigning batch IDs to new running requests. This leads to a **minor overhead of at most 1 step per request**: when a new request is assigned to the target batch, the verifier can only sample a single token since no draft is available. We consider this minor overhead **acceptable, since it may occur only at the initial step of a request, and all subsequent steps still benefit from parallelization**. Therefore, we plan to implement this strategy in our future version of MineDraft.
>
>
> > **Throughput-per-GPU comparison**
>
> Please kindly refer to our response to a similar question from Reviewer aL32 and Figure 26 at the anonymized link: https://anonymous.4open.science/r/anon-F221/fig26.jpg.
>
> > **Tail effect/fallback to standard SD**
>
> We appreciate the reviewer for questioning the impact of the tail effect in practice. We would like to clarify that the **tail effect occurs only when one batch is exhausted, which can only happen in offline benchmarking or testing** because the number of requests is fixed. In practice, LLM systems are usually deployed in online serving mode, where requests are unbounded. Therefore, the **tail effect is expected to never happen in real-world online serving deployment**. The fractions of total steps when standalone MineDraft operates in parallel vs. sequential (fallback) mode in our offline benchmarks are provided in Table 2 at this anonymized link: https://anonymous.4open.science/r/anon-F221/tbl2.jpg. We hope this quantified illustration of the tail effect can help address the limitation you pointed out.
>
> > **How does MineDraft handle variable output lengths?**
>
> We would like to thank the reviewer for raising the question on heterogeneous completion time handling. **MineDraft handles variable output lengths using continuous batching** (Lines 60 - 61, right column) -- the same way as vLLM does -- where the terminated request is replaced by a waiting request between execution steps by the Scheduler to saturate the batch. Since a normally finished request must be in target batch of previous step, a target batch will alternate to draft batch in the next step, and new running requests are always assigned to the draft batch of the next step, these mechanisms altogether keep the two batches full and balanced if the terminated request is finished normally (Lines 204 - 210, right column). Therefore, **batch balance is preserved provided that requests finish normally**. You may also refer to the example timeline illustration we provided in Sec. C.2 (Lines 702 - 707) and Fig. 7 to help you understand how MineDraft handles the cases when certain requests in a batch finish earlier than the rest of the batch.
>
> > **How sensitive is Thm 1 to the parametric form f(t)=1-e^{-αt}?**
>
> The exponential decay function aligns well with the practical behavior and is supported by our empirical results. As the number of tokens is essentially a discrete number, it is difficult to devise a perfect-fitting form, let alone analyze it. The exponential parametric form is the best choice, offering a decent fit while remaining analytically feasible.
>
> *We sincerely hope our responses, including the additional tables and figures, have addressed your concerns. We would deeply appreciate any further suggestions and feedback from you, as they have already helped improve our paper, particularly your advice on providing normalized performance comparisons in our experiments. If our answers have resolved the weaknesses and questions you raised, we would be truly grateful for your consideration in adjusting your scores accordingly.*

---

### Official Review · Reviewer_aL32 · 2026-03-13

**Soundness:** 4
**Presentation:** 4
**Significance:** 4
**Originality:** 4
**Overall Recommendation:** 5
**Confidence:** 4

**Summary:**

This paper proposes MINEDRAFT, a batch parallel speculative decoding framework that effectively hides drafting latency by overlapping the traditionally sequential drafting and verification stages of large language model inference through the alternating processing of two batches of requests.
Experimental results demonstrate that this system significantly accelerates model inference, achieving up to a 75% improvement in throughput and a 39% reduction in end-to-end latency compared to standard methods, and it has been successfully integrated as a highly practical, production-ready vLLM plugin.

**Compliance With Llm Reviewing Policy:**

Affirmed.

**Key Questions For Authors:**

1. The paper mentions that MINEDRAFT is implemented "at the cost of only 1 additional GPU." When benchmarking against Standard Speculative Decoding, was the total number of GPUs strictly aligned?
2. If deployed on a standard PCIe 4.0 server without high-speed NVLink, would the communication latency incurred by transferring Draft Tokens and probability distributions offset the benefits gained from parallelization?

**Limitations:**

Yes.

**Strengths And Weaknesses:**

### Strengths
1. **Significant Performance Improvements**\
Through an innovative "batch parallel" mechanism, the framework successfully overlaps the draft generation and verification stages, effectively hiding the drafting latency. Experimental results demonstrate that compared to standard speculative decoding (SD), MINEDRAFT improves throughput by up to 75% and reduces end-to-end latency by up to 39%.
2. **Solid Theoretical Foundation**\
The paper provides not only engineering implementations but also rigorous theoretical analysis. It mathematically proves that under moderate assumptions, parallel SD can reduce end-to-end latency by at least 37% compared to standard SD.
3. **High Compatibility and Orthogonality**\
The framework is "agnostic" to specific draft generation strategies. This means it can be seamlessly integrated with existing advanced speculative decoding algorithms to achieve superimposed performance optimizations.
4. **High Engineering and Practical Value**\
The authors developed this framework as a plug-and-play plugin for vLLM, a mainstream production-grade inference engine. It natively supports continuous batching and ingeniously maintains compatibility with the PagedAttention memory management mechanism via a "Scheduler Patch", making it highly suitable for direct industrial deployment.

### Weaknesses
1. Unexpected situations in real-world applications, such as the lack of support for user aborts or system preemptions. (Mentioned in the Limitations section)
2. Lack of support for chunked prefill. (Mentioned in the FAQ section at the end)
3. An unavoidable issue: the cost of additional GPU resources.

---

> ### Author Rebuttal · Authors · 2026-03-31
>
> Thank you for your insightful review and for noting that MineDraft **"effectively hides drafting latency"** via our novel Batch Parallelism and our vLLM plugin is **"highly practical, production-ready"**. We also appreciate your recognition of our paper’s **"solid theoretical foundation"**, **"high compatibility and orthogonality with advanced speculative decoding algorithms"**, and **"high engineering and practical value"**. For clarity, we have grouped related weaknesses and questions and addressed them as follows:
>
> > **Support for user aborts or system preemptions and chunked prefill**
>
> We would like to clarify that, **technically, MineDraft supports user aborts and system preemptions because it is built on vLLM, which provides native support for these features**. What MineDraft lacks is the graceful handling of these cases. However, we have discussed a possible measure to resolve this issue in Sec. A (Lines 560 - 564): continuously applying the `assign` operation, as specified in Sec. 5.1 (Lines 194 - 203, right column) throughout the lifespan of MineDraft. Although this measure can introduce less efficient "idle" steps (when a new running request is assigned to the target batch, so no drafts are available for verification), they occur at most once per request, which we consider acceptable for the long-term performance benefits. Although chunked prefill is not currently supported in MineDraft, it is an interesting future direction (Lines 433 - 435, right column).
>
> > **Additional GPU resources.**
>
> Though MineDraft currently uses an additional GPU, this issue can be resolved by using the weight padding method as mentioned in Sec. 7 (Lines 435 - 438, right column). We view the extra dedicated drafter GPU as an **opportunity to support advanced and complex drafting strategies (discussed in Sec. 2, Lines 77 - 80 of right column) instead of a cost**, since such strategies usually require additional GPU time and resources. Moreover, we compared the performance of MineDraft with 5 GPUs to that of standard SD with 4 GPUs in our experiments, achieving a ~75% throughput gain and a ~39% latency reduction with 25% more GPU resources.
>
> > **Benchmarking when the total number of GPUs is strictly aligned?**
>
> Thank you for this suggestion. Given that we run MineDraft and PEARL with 5 GPUs and standard SD with 4 GPUs in our experiments, we normalize average throughput by dividing the average throughputs of MineDraft and PEARL by 5 and that of standard SD by 4. For end-to-end latency, we normalize it by multiplying the end-to-end latencies of MineDraft and PEARL by 1.25. **Please kindly refer to Figures 26 and 27 at the respective anonymized links:** https://anonymous.4open.science/r/anon-F221/fig26.jpg and https://anonymous.4open.science/r/anon-F221/fig27.jpg, **which show normalized throughput and latency comparisons.** Figures 26 and 27 show that MineDraft outperforms standard SD and PEARL except when $k$ is small (between 1 and 2), where MineDraft may underperform compared to standard SD. **Overall, MineDraft improves normalized average throughput by ~40.55% and end-to-end latency by ~24.38% over standard SD.** We hope this could help disambiguate our benchmark comparisons about which the reviewer is concerned.
>
> > **Server configuration (PCIe 4.0 vs NVLink)**
>
> Thank the reviewer for raising the valuable concern about parallelization benefits, given the server's standard PCIe 4.0 configuration. We would like to clarify that **our experiments are all run on standard PCIe 4.0 servers without NVLink or other direct GPU-to-GPU communication medium**. Our experiments have shown that the performance benefits of MineDraft would not be affected in the absence of a high-speed GPU-to-GPU communication medium, and we expect such technologies would further amplify MineDraft's performance gains, since they significantly reduce GPU-to-GPU communication latency and improve unidirectional throughput. Table R1 shows the average communication latencies profiled on Settings 1-3 and the Spec-Bench dataset using Nsight Systems, along with the corresponding estimated communication latencies after upgrading to NVLink 4.
>
> Table R1. Comparison of average PCIe 4.0 communication latencies and estimated NVLink 4 latencies on Settings 1-3 and Spec-Bench dataset. Empirical PCIe 4.0 communication latencies are profiled using Nsight Systems.
>
> |Setting|$k$|PCIe 4.0 latency (ms)|Estimated NVLink 4 latency (ms)|
> |---|---|---|---|
> |Qwen3 32B-0.6B (Setting 1)|3|8.857|0.470
> ||4|9.438|0.491
> |Qwen3 32B-1.7B (Setting 2)|3|8.554|0.453
> ||4|9.711|0.513
> |Qwen3 32B-4B (Setting 3)|3|7.865|0.417
> ||4|9.026|0.477
>
> *We sincerely hope our response has addressed your concerns. We are also grateful for any further questions or comments you may have; if you have any additional concerns, please let us know. If our answers have helped you better understand our paper and resolved your questions, we would deeply appreciate your openness to adjust your scores accordingly.*

---

### Official Review · Reviewer_nuQ1 · 2026-03-21

**Soundness:** 3
**Presentation:** 3
**Significance:** 3
**Originality:** 3
**Overall Recommendation:** 4
**Confidence:** 4

**Summary:**

This paper introduces minedraft, which is a batch parallel speculative decoding framework for effectively hiding drafting latency via draft-target model overlapping. It is proven that this PSD can be much more efficient than traditional SD. The results shows that minedraft can improve throughput and latency by up to 75% and 39% compared with standard SD.

**Compliance With Llm Reviewing Policy:**

Affirmed.

**Key Questions For Authors:**

See the limitations section.

**Limitations:**

Weakness:
1. While the paper presents a concrete batched systems design, the high-level idea of parallel speculative decoding and overlapping drafting with verification appears related to prior work, so the true novelty may lie more in the specific serving implementation than in the core concept itself.
2. since the method splits a large batch into two smaller batches, it is unclear whether the paper sufficiently isolates the throughput loss from reduced batch efficiency versus the gain from overlapping drafting and verification.

**Strengths And Weaknesses:**

1. This paper proposes an interesting and novel systems idea that splits a large batch into two smaller batches to form a pipeline, allowing drafting and verification to overlap and thereby hide drafting latency.

2. The experiments are quite comprehensive, covering different model sizes, GPU setups, and draft models, and the method is implemented and evaluated on top of vLLM, which strengthens its practical relevance.

---

> ### Author Rebuttal · Authors · 2026-03-31
>
> We would like to thank the reviewer for taking the time to carefully review our paper. We appreciate your noting that we have proved MineDraft **"can be much more efficient than traditional SD"** and have shown in our empirical results that MineDraft **"can improve throughput and latency by up to 75% and 39% compared with standard SD"**. We are glad for your recognition that our paper **"proposes an interesting and novel systems idea"** and our **"experiments are quite comprehensive"**. Meanwhile, we have addressed the weakness that you raised and answered your question as follows:
>
> > **While the paper presents a concrete batched systems design, the high-level idea of parallel speculative decoding and overlapping drafting with verification appears related to prior work, so the true novelty may lie more in the specific serving implementation than in the core concept itself.**
>
> We would like to thank the reviewer for raising the concern about novelty. The serving implementation of parallel speculative decoding (PSD) *is* the core contribution, but it is technically non-trivial. While the high-level idea of overlapping drafting and verification is natural, no prior work has realized it efficiently in the *batch-request* setting. Specifically, PEARL (Liu et al., 2025a) explicitly cannot scale to multi-request batches (Lines 32 - 46 of the right column of our paper: "scaling this method to batched requests remains challenging"); Wang et al. (2024) and Timor et al. (2025) require substantially increased VRAM; and Xiao et al. (2024) require a dedicated training stage. MineDraft's Batch Parallelism paradigm is the first to achieve PSD with only 1 additional GPU, no retraining, and full compatibility with production vLLM (continuous batching, PagedAttention). The KV block management patch (Sec. 5.3), batch state alternation, and fallback mechanisms are each *necessary* for correctness in a production serving stack; none are trivially derived from the high-level concept. At a high level, **MineDraft alternatively drafts and verifies two batches to overlap drafting with verification** (introduced in Sec. 5, Lines 191 - 197 of the left column), thereby minimizing the idle period of the verifier and the requirements of VRAM and additional GPUs. Moreover, MineDraft neither introduces verification failures (illustrated in Figure 28 at this anonymized link: https://anonymous.4open.science/r/anon-F221/fig28.jpg) nor saturates the dedicated drafter GPU, making it well-suited for integration with advanced drafting methods (evaluated in Sec. 6.2, Lines 375 - 383 of the left column and Line 371 of the right column). Lastly, we provided a theoretical analysis under mild assumptions that validates the efficiency of MineDraft (Sections 4 and B) and a vLLM plugin implementation that supports hassle-free deployment (included in the Supplementary Material), pushing parallel speculative decoding to the production-ready stage.
>
> > **since the method splits a large batch into two smaller batches, it is unclear whether the paper sufficiently isolates the throughput loss from reduced batch efficiency versus the gain from overlapping drafting and verification.**
>
> We would like to clarify that, **MineDraft maintains two batches of size $m$, rather than splitting a batch of size $m$ into two smaller batches** (specified in Sec. 5, Lines 191 - 197 of left column), i.e., the "batch efficiency" is not affected. The total token capacity $C$ in our experiment is fixed across both batches, so each sub-batch processes *m* requests per step, identical to standard SD. The performance improvement is therefore attributable purely to parallelism.
>
> Our parallelism design allows us to:
> 1. saturate the verifier GPUs while saving time on the drafter GPU for more complicated drafting strategies/methods (illustrated in Figure 1 of the paper);
> 2. alleviate the memory contention problem in standard SD (discussed in Sec. 6.2, Lines 372 - 383 of the right column and Lines 402 - 403 of the left column); and
> 3. achieve up to 75.68% throughput gain (Lines 315 - 316, right column) and 39.51% end-to-end latency reduction compared to standard SD (Lines 874 - 875).
>
> *We sincerely hope our response has addressed your concerns. We would appreciate further feedback from you if you have any additional comments or questions that we missed in our response. If our answers helped clarify our work and resolve your questions, we would be deeply grateful for your consideration in adjusting your scores accordingly.*

---

> > ### Author Rebuttal · Reviewer_nuQ1 · 2026-04-07
> >
> > I think the rebuttal makes sense and I maintain positive for this paper.

---

> > > ### Author Response · Authors · 2026-04-08
> > >
> > > Thank you for your time and careful review of our paper. Your feedback will help us further improve the clarity of our paper.
> > >
> > > Given that your concerns have now been fully addressed, we would greatly appreciate your consideration in revisiting your *Overall Recommendation* to reflect these clarifications and supporting our paper in the subsequent discussion.

---

### Decision · Program_Chairs · 2026-04-30

**Decision:**

Accept (regular)

**Comment:**

The paper introduces MineDraft, a framework designed to optimize speculative decoding (SD) by transitioning from sequential to parallel execution. While standard SD forces the target model to wait for the draft model to finish, MineDraft utilizes Batch Parallelism. By maintaining two alternating batches of requests and dedicated GPU resources for drafting, the system overlaps the drafting phase of one batch with the verification phase of another. This allows the system to effectively "hide" drafting latency. The authors provide both a theoretical foundation (proving a minimum $37\%$ latency reduction under specific conditions) and a practical, production-ready implementation as a vLLM plugin.

The paper takes a known bottleneck and applies a clean, intuitive, and well-executed architectural change to solve it. All reviewers had positive scores (4s and 5s) following the rebuttal, citing that the authors' clarifications and additional data fully resolved their concerns. The work is technically sound, highly relevant to the current LLM landscape, and ready for deployment.